# The interplay of stiffness and force anisotropies drives embryo elongation

**Thanh Thi Kim Vuong-Brender[1,2]\*, Martine Ben Amar[3,4], Julien Pontabry[2†], Michel Labouesse[1,2]\***

[1]Laboratoire de Biologie du Développement - Institut de Biologie Paris Seine (LBD - IBPS), Sorbonne Universités, UPMC Univ Paris 06, CNRS, Paris, France; [2]Development and Stem Cells Program, IGBMC, CNRS (UMR7104), INSERM (U964), Université de Strasbourg, Illkirch, France; [3]Laboratoire de Physique Statistique, Ecole Normale Supérieure, UPMC Université Pierre et Marie Curie, Université Paris Diderot, CNRS, Paris, France; [4]Institut Universitaire de Cancérologie, Faculté de Médecine, Université Pierre et Marie Curie-Paris, Paris, France

**Abstract** The morphogenesis of tissues, like the deformation of an object, results from the interplay between their material properties and the mechanical forces exerted on them. The importance of mechanical forces in influencing cell behaviour is widely recognized, whereas the importance of tissue material properties, in particular stiffness, has received much less attention. Using *Caenorhabditis elegans* as a model, we examine how both aspects contribute to embryonic elongation. Measuring the opening shape of the epidermal actin cortex after laser nano-ablation, we assess the spatiotemporal changes of actomyosin-dependent force and stiffness along the antero-posterior and dorso-ventral axis. Experimental data and analytical modelling show that myosin-II-dependent force anisotropy within the lateral epidermis, and stiffness anisotropy within the fiber-reinforced dorso-ventral epidermis are critical in driving embryonic elongation. Together, our results establish a quantitative link between cortical tension, material properties and morphogenesis of an entire embryo.

**\*For correspondence:**
vuongthikimthanh@gmail.com
(TTKV-B); michel.labouesse@
upmc.fr (ML)

**Present address:** †Helmholtz Zentrum, Institute of Epigenetics and Stem Cells, München, Germany

**Competing interests:** The authors declare that no competing interests exist.

## Introduction

Morphogenesis and organ formation rely on force distribution and tissue material properties, which are often heterogeneous and evolve over time. Forces are generated through a group of relatively well-conserved molecular motors associated with the cytoskeleton, among which, myosin II linked to actin filaments is the most prevalent during epithelial morphogenesis (*Vicente-Manzanares et al., 2009*). The spatial distribution and dynamics of myosin II greatly influence morphogenetic processes (*Levayer and Lecuit, 2012*). In particular, the asymmetric distribution of the actomyosin network and its pulsatile behaviour define the direction of extension during *Drosophila* germband elongation (*Bertet et al., 2004*; *Blankenship et al., 2006*), *Drosophila* renal tubule formation (*Saxena et al., 2014*) or *Xenopus* mesoderm convergent extension (*Shindo and Wallingford, 2014*). The implications of mechanical forces on cell behavior have been intensively investigated (*Zhang and Labouesse, 2012*; *Heisenberg and Bellaïche, 2013*), but many fewer studies have considered the impact of tissue material properties *in vivo*, except for their influence on cell behaviour *in vitro* (*Kasza, 2007*).

Embryonic elongation in *C. elegans* represents an attractive model for studying morphogenesis, as it offers single-cell resolution and powerful genetic analysis. During its elongation, the embryo evolves from a lima-bean shape to a typical cylindrical shape with a four-fold increase in length, without cell migration, cell division, or a notable change in embryonic volume

**eLife digest** Animals come in all shapes and size, from ants to elephants. In all cases, the tissues and organs in the animal's body acquire their shape as the animal develops. Cells in developing tissues squeeze themselves or push and pull on one another, and the resulting forces generate the final shape. This process is called morphogenesis and it is often studied in a worm called *Caenorhabditis elegans*. This worm's simplicity makes it easy to work with in the laboratory. Yet processes that occur in *C. elegans* also take place in other animals, including humans, and so the discoveries made using this worm can have far-reaching implications.

As they develop, the embryos of *C. elegans* transform from a bean-shaped cluster of cells into the characteristic long shape of a worm, with the head at one end and the tail at the other. The force required to power this elongation is provided by the outer layer of cells of the embryo, known as the epidermis. In these cells, motor-like proteins called myosins pull against a mesh-like scaffold within the cell called the actin cytoskeleton; this pulling is thought to squeeze the embryo all around and cause it to grow longer.

Six strips of cells, running from the head to the tail, make up the epidermis of a *C. elegans* embryo. Myosin is mostly active in two strips of cells that run along the two sides of the embryo. In the strips above and below these strips (in other words, those on the upper and lower sides of the worm), the myosins are much less active. However, it is not fully understood how this distribution of myosin causes worms to elongate only along the head-to-tail axis.

Vuong-Brender et al. have now mapped the forces exerted in the cells of the worm's epidermis. The experiments show that, in the strips of cells on the sides of the embryo, myosin's activity causes the epidermis to constrict around the embryo, akin to a boa constrictor tightening around its prey. At the same time, the actin filaments in the other strips form rigid bundles oriented along the circumference that stiffen the cells in these strips. This prevents the constriction from causing the embryo to inflate at the top and bottom strips. As such, the only direction the embryo can expand is along the axis that runs from its head to its tail.

Together, these findings suggest that a combination of oriented force and stiffness ensure that the embryo only elongates along the head-to-tail axis. The next step is to understand how this orientation and the coordination between cells are controlled at the molecular level.

(*Sulston et al., 1983*; *Priess and Hirsh, 1986*) (*Figure 1a*). This process requires the epidermal acto-myosin cytoskeleton, which acts mostly in the lateral epidermis (also called seam cells), while the dorso-ventral (DV) epidermal cells may remain passive (Appendix 1) (*Wissmann et al., 1997*; *1999*; *Shelton et al., 1999*; *Piekny et al., 2003*; *Diogon et al., 2007*; *Gally et al., 2009*; *Chan et al., 2015*; *Vuong-Brender et al., 2016*). Indeed, the non-muscle myosin II is concentrated in seam cells; in addition, short disorganized actin filaments, which favour actomyosin contractility, are present in seam cells but not in the DV epidermis, where they instead form parallel circumferential bundles (*Figure 1b–d*) (*Gally et al., 2009*; *Priess and Hirsh, 1986*). The actomyosin forces are thought to squeeze the embryo circumferentially, thereby increasing the hydrostatic pressure and promoting embryo elongation in the antero-posterior (AP) direction (*Priess and Hirsh, 1986*) (*Figure 1e*).

Although the published data clearly implicate myosin II in driving elongation, they raise a number of issues. First, myosin II does not show a polarized distribution (*Figure 1f,g*), nor does it display dynamic pulsatile foci at this stage; hence, it is difficult to account for the circumferential squeezing. Moreover, force measurements are lacking to establish that the actomyosin network does squeeze the embryo circumferentially. Second, a mechanical continuum model is needed to explain how the embryo extends preferentially in the AP direction.

To address those issues, we used laser ablation to map the distribution of mechanical stress (i.e the force per unit area) and to assess tissue stiffness (i.e. the extent to which the tissue resists deformation) in the embryonic epidermis. We then correlated the global embryonic morphological changes with these physical parameters. Finally, we developed continuum mechanical models to account for the morphological changes. Altogether, our data and modelling demonstrate that the

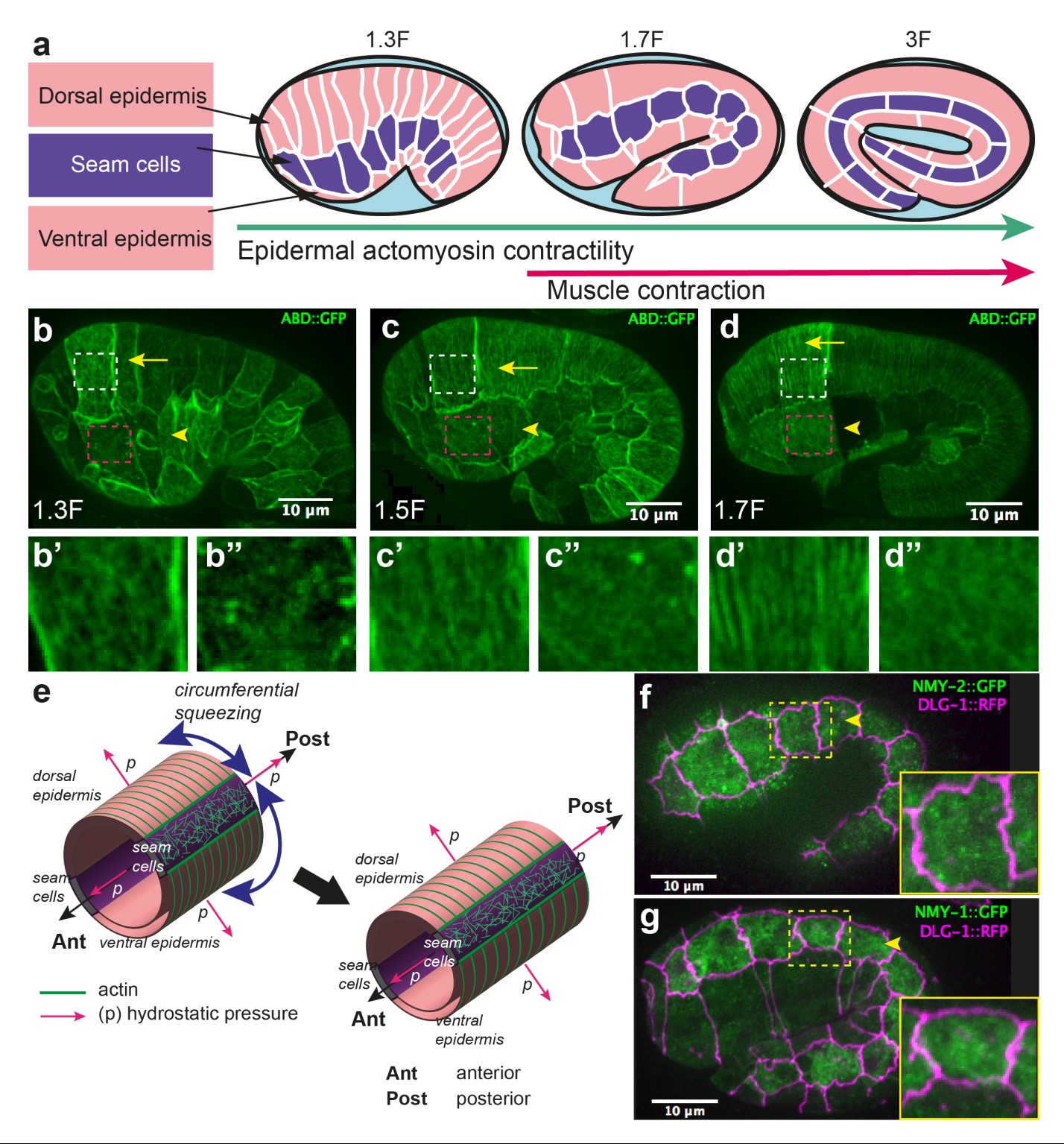

**Figure 1.** Overview of *C. elegans* embryonic elongation. (**a**) Embryonic elongation in *C. elegans* is driven in part by epidermal actomyosin contractility and in part by muscle contractions. The length of the embryo is used for staging: 2-fold (2F) stage means roughly 2-fold increase in length from the beginning of elongation. Representative stages are shown; anterior to the left, dorsal up. (**b, c, d**) Actin filament organization at the 1.3F, 1.5F and 1.7F stages, respectively, visualized with an ABD::GFP marker. Actin filaments progressively organize into circumferential parallel bundles in DV cells (arrows), arrowheads point to seam cells. Note that the integrated ABD::GFP marker shows some cell to cell variation in expression. (**b', b", c', c", d', d"**) Close-up images of actin pattern in DV cells (from the area in the white rectangle) and seam cells (from the area in the pink rectangle), respectively, of the

*Figure 1 continued on next page*

*Figure 1 continued*

images in (**b**), (**c**) and (**d**) respectively. (**e**) Actomyosin forces squeeze the embryo circumferentially to make it elongate in the antero-posterior direction. (**f, g**) Endogenous distribution of the two non-muscle myosin II isoforms visualized with the CRISPR GFP-labelled myosin heavy chains NMY-2 (**f**) and NMY-1 (**g**). Arrowheads point to seam cells, which are delineated by the junctional marker DLG-1::RFP.

distribution of forces in the seam cells and the stiffness in the DV epidermis must be polarized along the circumferential axis (or DV axis) to drive elongation.

## Results

### Measuring the mechanical stress on the actin cortex through laser ablation

To measure the stress distribution on the actin cortex, we used laser nano-ablation, which has now become a standard method to assess forces exerted in cells, to sever the actin cytoskeleton and to observe the shape of the opening hole (*Figure 2a*). We visualized actin with a GFP- or mCherry-labelled actin-binding-domain protein (ABD) expressed in the epidermis (*Gally et al., 2009*)

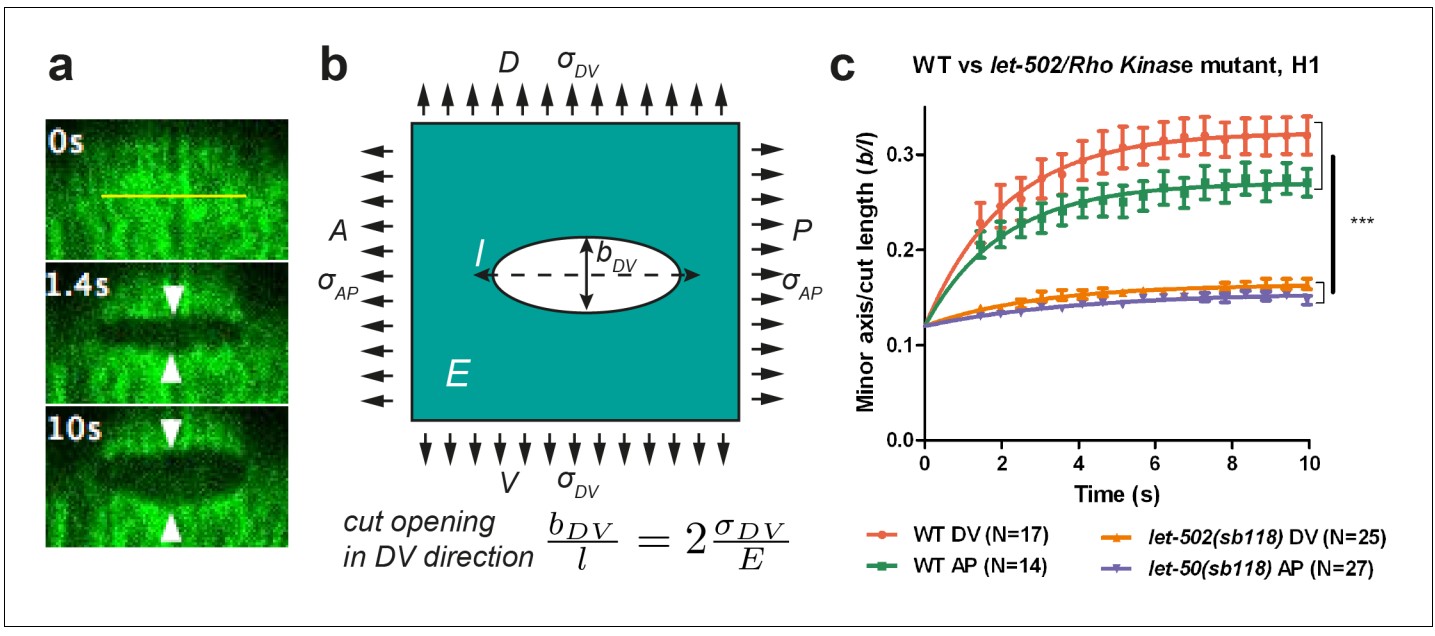

**Figure 2.** Physical model using the shape of the cut opening at equilibrium to measure the ratio of stress to Young modulus. (**a**) The GFP-labelled actin cortex of HYP7 dorsal epidermal cell at the 1.7F stage before (0 s), 1.4 s and 10 s after laser severing; the cut made along the AP direction and was 5 μm in length. Double arrowheads indicate the distance between cut borders, which increases with time. (**b**) Model of epidermal cells as an infinite elastic plane under biaxial stress in the AP and DV directions after an incision of length $l$. The final shape of the cut opening is an ellipse. The cut opening in the DV direction (after an incision along the AP direction) depends on the stress along the DV direction and the Young modulus (see text). (**c**) The opening depends on myosin II activity: comparison of the cut response in the seam cell H1 between wild-type (WT) and *let-502(sb118)/Rho kinase* mutant embryos at a stage when muscles start to twitch (around 1.5F). The average value and standard error are reported. Time zero, moment of the cut; DV and AP show direction of opening. Two-tailed t-test, ***p=$4*10^{-7}$ between WT DV and *let-502(sb118)* DV, p=$4*10^{-6}$ between WT AP and *let-502(sb118)* AP; N, number of embryos examined.

The following figure supplements are available for figure 2:

**Figure supplement 1.** Evolution of the distance between the cut borders (the minor axis of the cut opening) versus time.

**Figure supplement 2.** Calcium imaging of ablated embryos.

(*Figure 1b–d*). We adjusted the region of interest to cut within one cell, restricting our analysis to the early phase of elongation (≤1.7F; for staging, see *Figure 1* legend).

We observed two types of ablation responses (see Materials and methods). In the first (accounting for >80% of the cases), the opening hole within the actin cytoskeleton reached equilibrium in less than 10 s, and resealed within less than 2 min (*Figure 2—figure supplement 1*, *Video 1*). In such embryos, actin occasionally accumulated around the cut borders but not around cell borders (*Video 1*). Imaging calcium levels, which can rise after laser wounding (*Xu and Chisholm, 2011*; *Razzell et al., 2013*; *Antunes et al., 2013*), showed either no change or a localized increase (*Video 2*, *Figure 2—figure supplement 2a,b*). In the second ablation reponse, an actin ring accumulated around the cell borders during the repair process (*Video 3*) and a calcium wave propagated to nearby epidermal cells (*Video 4*; *Figure 2—figure supplement 2c*). Embryos showing the first response continued to develop and hatched, whereas those showing the second response arrested their development and eventually died. In all subsequent studies, we only took into account the first type of response, which should correspond to a local cortex disruption.

To compare the response between different conditions, we detected the cut-opening shape, which we fitted with an ellipse to derive the shape parameters (see Materials and methods). The laser setup we used did not enable us to image the recoil dynamics within the first second after the cut, which other investigators previously used to assess the extent of mechanical stress (*Rauzi and Lenne, 2015*; *Smutny et al., 2015*; *Saha et al., 2016*). To circumvent this issue, we developed a novel analysis method to derive mechanical stress data, based on the equilibrium shape of a thin cut in an infinite elastic isotropic plane, subjected to biaxial loading (stress applied in two perpendicular directions) (*Theocaris, 1986*). The rationale for approximating the epidermis to such a plane is further outlined in the Appendices 2 and 3. In these conditions, a thin cut will open to form an elliptical hole at equilibrium (*Figure 2a*). The opening of the cut reflects mechanical stress in the direction perpendicular to the cut direction.

We cut the epidermal actin specifically in the AP and DV directions, which we found to correspond to the stress loading directions (*Figure 2b*; Appendix 2). For a cut in the AP direction, the minor axis of the ellipse at equilibrium, $b_{DV}$, will be proportional to the cut length, $l$, and to the ratio of stress in the DV direction, $\sigma_{DV}$, over the Young modulus $E$ of the plane (*Theocaris et al., 1986*) (*Figure 2b*):

$$b_{DV} = 2\frac{\sigma_{DV}}{E}l \qquad (1)$$

We will call the ratio $\frac{b_{DV}}{l}$ the opening in the DV direction of a cut made along the AP direction (*Figure 2b*), and similarly $\frac{b_{AP}}{l}$ the opening in the AP direction of a cut made in the DV direction. Thus, we used the opening of the hole in a given direction to derive the stress in that direction.

We compared the conclusions drawn from this method with methods relying on the recoil dynamics (*Rauzi and Lenne, 2015*; *Smutny et al., 2015*; *Saha et al., 2016*) (Appendix 3). The half-time of the cut border relaxation, which depends on the ratio of viscosity over stiffness, was similar in the AP and DV directions (Appendix 3), supporting the hypothesis that the seam cell cortex is isotropic. We found an agreement between both methods for the AP versus DV stress ratio, and similar trends for the stress magnitude.

To further examine the validity of this method, we performed two tests. First, the theory described above (*Theocaris, 1986*) predicts that the minor to major axis ratio of the opening

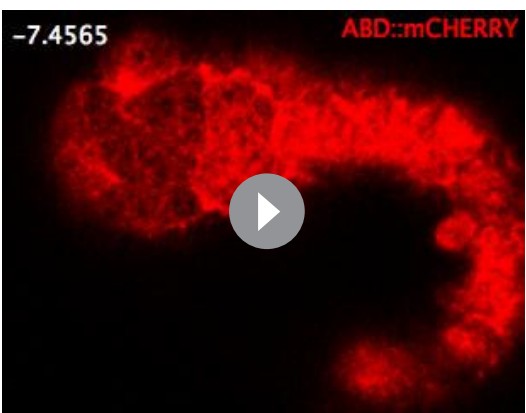

**Video 1.** Local disruption of actin cortex with laser ablation, visualized with the actin marker (ABD:: mCHERRY) expressed under the epidermal promoter *lin-26*. 0 s time corresponds to the first picture after the laser cut. The yellow line shows the cut region.

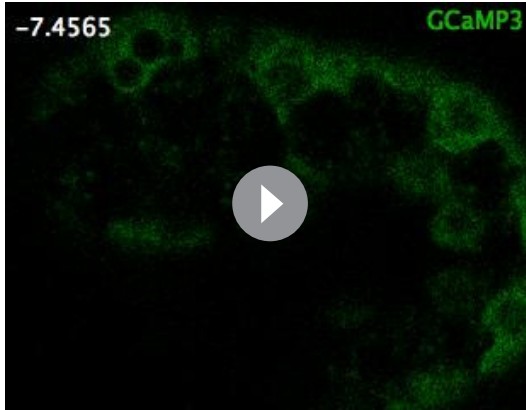

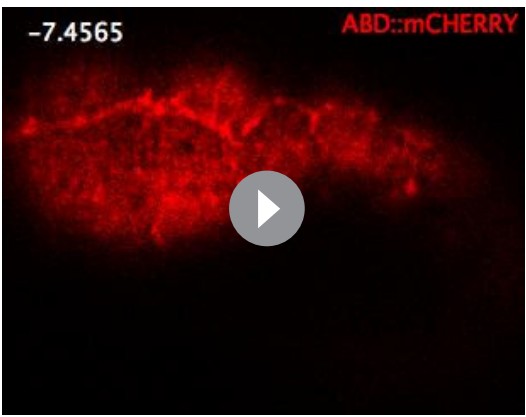

**Video 2.** Local disruption of actin cortex with laser ablation does not induce noticeable change in calcium level. The calcium sensor GCaMP3 was expressed under the epidermal promoter *dpy-7* (which provides strong expression in the dorso-ventral cells). 0 s time corresponds to the first picture after the laser cut. The yellow line shows the cut region. This is the same embryo as that shown in *Video 1*.

**Video 3.** Wound-healing response after laser ablation visualized with the actin marker ABD::mCHERRY expressed under the epidermal promoter *lin-26*. 0 s time corresponds to the first picture after the laser cut. The yellow line shows the cut region.

ellipse is independent of the initial cut length. We found that it is the case when the cut length varied from 3 µm to 6 µm (Appendix 4). Second, to prove that the opening observed after laser cutting depends on myosin II activity, we performed cuts in embryos defective for the main myosin II regulator, LET-502/Rho-kinase (*Gally et al., 2009*). As shown in *Figure 2c*, the opening in the seam cell H1 at the 1.5F stage in *let-502(sb118ts)* embryos changed very little and was significantly smaller than that in WT embryos, consistent with a decrease in mechanical stress.

Thus, we feel confident that the method based on the opening shape measures actomyosin-dependent stress and can be used reliably to report on stress differences along the DV and AP directions.

## Stress anisotropy in seam cells correlates with embryonic morphological changes

We applied the method described above on three seam cells (head H1, body V3, tail V6; *Figure 3a*) because myosin II acts mainly in seam cells (*Gally et al., 2009*), and compared the response with embryonic morphological changes. We focused on the anisotropy of stress between the DV and AP directions (difference of stress along both directions) in a given cell (*Figure 3b*). Indeed, in other systems, such as *Drosophila* embryos (*Rauzi et al., 2008*) and *C. elegans* zygotes (*Mayer et al., 2010*), this parameter is critical.

At the 1.3F stage in H1, there was no significant stress anisotropy; however, as the embryo elongated to the 1.5F and 1.7F stages, the stress became anisotropic (*Figure 3b*). In V3, the anisotropy of stress evolved in the opposite direction, with higher stress anisotropy at the 1.3F stage compared to the 1.5F stage (*Figure 3b*). In V6, the stress was slightly

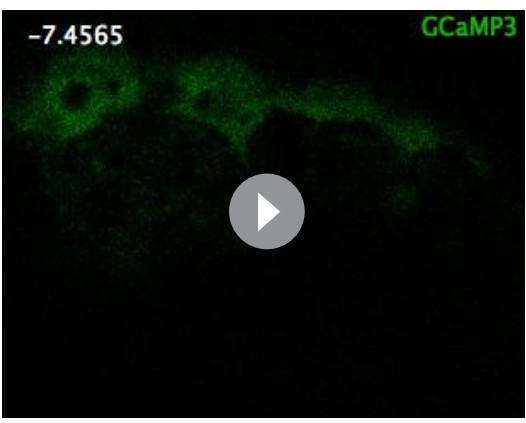

**Video 4.** Calcium wave propagation in a wound-healing response after laser ablation. The calcium sensor GCaMP3 was expressed under the epidermal *dpy-7* promoter (which provides strong expression in the dorso-ventral cells). 0 s time corresponds to the first picture after the laser cut. The yellow line shows the cut region. This is the same embryo as that shown in *Video 3*.

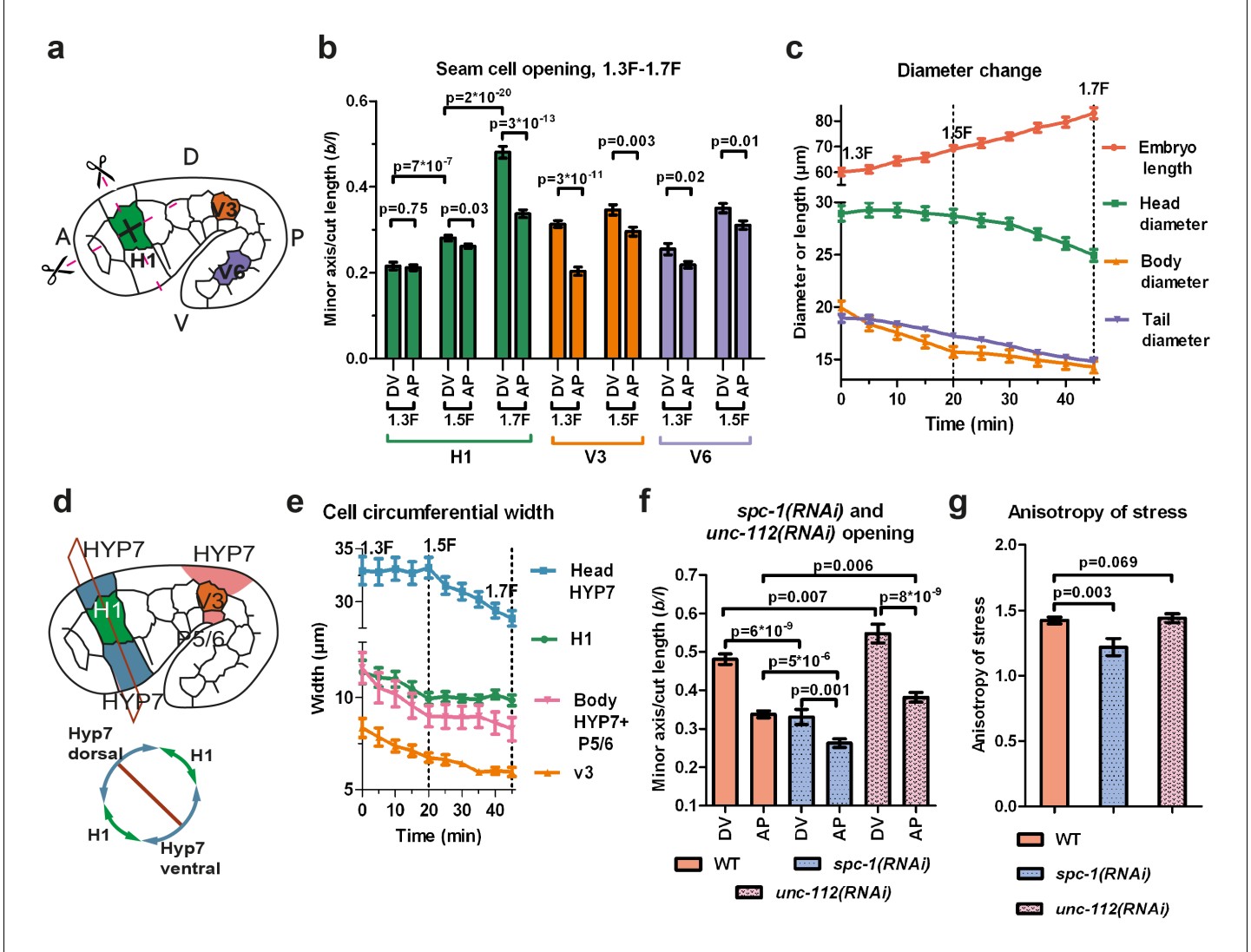

**Figure 3.** Stress anisotropy in seam cells correlates with morphological changes and partially depends on the spectrin cytoskeleton. (a) Scheme showing laser ablation experiments in the AP and DV directions for H1, V3 and V6 seam cells at different stages. A, anterior; P, posterior; D, dorsal; V, ventral. (b) Cut opening in H1, V3 and V6 from the 1.3F to the 1.7F stages (see *Figure 2b*). The p-values from two-tailed t-tests are reported. (c) Changes in embryo length, head diameter at the level of H1, body diameter at the level of V3 and tail diameter at the level of V6, between the 1.3F and 1.7F stages. N = 10. (d) Scheme showing the measurement of circumferential cell width in the head (above) and corresponding section (below). (e) The circumferential width of H1, V3, head and body DV cells (averaged for dorsal and ventral cells). N = 10. (f) Measures of the cut opening in H1 for WT, *spc-1(RNAi)* treated and *unc-112(RNAi)* muscle-defective embryos at a stage equivalent to the 1.7F stage. The p-values of two-tailed t-tests are reported. (g) Comparison of the stress anisotropy in H1, defined by DV/AP stress, between WT 1.7F stage, *spc-1(RNAi)* embryos and *unc-112(RNAi)* embryos at the equivalent 1.7F stage. The p-values of Z-tests are reported. The number of embryos used for ablation is given in *Supplementary file 1*. For (b,c,e–g), the average (or calculated) values and standard errors are reported.

The following figure supplement is available for figure 3:

**Figure supplement 1.** Elongation curves normalized to the initial embryo length for different genetic backgrounds.

anisotropic at both the 1.3F and the 1.5F stage (*Figure 3b*). In all cells, whenever the stress became anisotropic, it was higher in the DV direction. Overall, the opening increased as the embryo elongated from the 1.3F to the 1.5F stage and from the 1.5F to the 1.7F stage for H1.

To correlate the stress anisotropy with the morphological changes in the embryo, we used markers labelling cortical actin (an ABD) and junctions (HMR-1/E-cadherin). We observed that the

head, body and tail diameter (at the levels of H1, V3 and V6, respectively) decreased at different rates over time (*Figure 3c*), as also observed by Martin and colleagues (*Martin et al., 2014*). The head diameter did not diminish between the 1.3F and 1.5F stages when the stress was nearly isotropic, but decreased significantly between the 1.5F and 1.7F stages as the stress anisotropy increased. Conversely, the body diameter decreased most rapidly between the 1.3F and 1.5F stages, when the stress was highly anisotropic, then changed at a lower pace beyond the 1.5F stage, when the stress became less anisotropic. Finally, the tail diameter decreased nearly linearly between the 1.3F and 1.7F stages, at a lower rate than the body diameter, coinciding with a smaller anisotropic stress in V6. Thus, the local morphological changes within the embryo correlate with locally higher stress in the DV compared to the AP direction.

To define whether all cells contribute equally to the diameter change, we quantified the circumferential width of the epidermal cells H1, V3 and their adjacent DV cells (*Figure 3d,e*). At the level of V3, the decrease in body diameter came from both seam (V3) and DV cells, whereas in the head, it came mainly from DV cells (*Figure 3e*). Collectively, our results strongly suggest that stress anisotropy correlates with morphological changes. Furthermore, we found that both seam and DV epidermal cells contribute to the changes in embryo diameter, irrespective of their level of active myosin II.

## The establishment of stress anisotropy depends in part on the spectrin cytoskeleton

Taking the H1 cell as an example, we considered some cellular factors that could contribute to the stress anisotropy in seam cells: (i) actin-anchoring proteins, and (ii) muscle-induced tension. To ease comparisons, we defined the anisotropy of stress (AS) as

$$AS = \frac{DV\ stress}{AP\ stress} = \frac{\sigma_{DV}}{\sigma_{AP}} \tag{2}$$

which can be derived from the ratio of the opening along the DV and AP directions, see *Equation (1)*.

First, we examined the actin-anchoring spectrin cytoskeleton, which is essential for embryonic elongation (*Moorthy et al. 2000*; *Norman and Moerman, 2002*; *Praitis et al., 2005*). In *spc-1 (RNAi)* embryos, at a developmental timing equivalent to the 1.7F stage in control embryos, we found a smaller opening in both AP and DV directions and a decrease of AS compared to WT (*Figure 3f,g*). This may account for the slower elongation rate of *spc-1(RNAi)* embryos and their arrest at the 2F stage (*Figure 3—figure supplement 1*). Thus, spectrin partially contributes to the AS at the 1.7F stage.

Second, we wondered whether muscle contractions, which start after the 1.5F stage, could account for AS changes (*Figure 3b*). Compared to controls, embryos that are depleted in UNC-112/Kindlin, which mediates sarcomere assembly (*Rogalski et al., 2000*), showed a significantly larger opening in both the AP and DV directions at the 1.7F stage, but no change in stress anisotropy (*Figure 3f,g*). This is consistent with their wild-type elongation rate up to the 2F stage (*Figure 3—figure supplement 1*). Thus, AS establishment in the H1 cell after the 1.5F stage is independent of muscle contractions.

## A mechanical model for seam cell elongation depending on stress anisotropy

To define the possible causal relationship between the AS and embryonic shape changes, we aimed to simplify the shape of the embryo in order to allow the application of classical physical laws such as the Young-Laplace equation, which predicts the relationship between surface tension and the surface curvature. As illustrated in *Figure 1*, the embryo has a circular section and a cylindrical or conical shape depending on the stage, in which the epidermis is relatively thin (100 nm to 2 μm, depending on areas; www.wormatlas.org) when compared to the embryo diameter (25 μm). Within the embryo, the epidermis is subjected to hydrostatic pressure when the section decreases (*Priess and Hirsh, 1986*). We can thus model the *C. elegans* embryo as an isotropic thin-wall (the epidermis) vessel with capped ends under hydrostatic pressure, and can determine the relationship between the mechanical stress on the epidermis and the embryo shape.

First, we calculated the anisotropy of stress on the wall of such a vessel. For an axisymmetric vessel, the AS on the wall depends on the surface curvature and the radius (Appendix 5), which for simple geometrical configurations can be written as shown in *Figure 4a–c*. Typically, the AS factor, or the DV to AP stress ratio, is equal to one for a sphere, equal to two for a cylinder and takes an

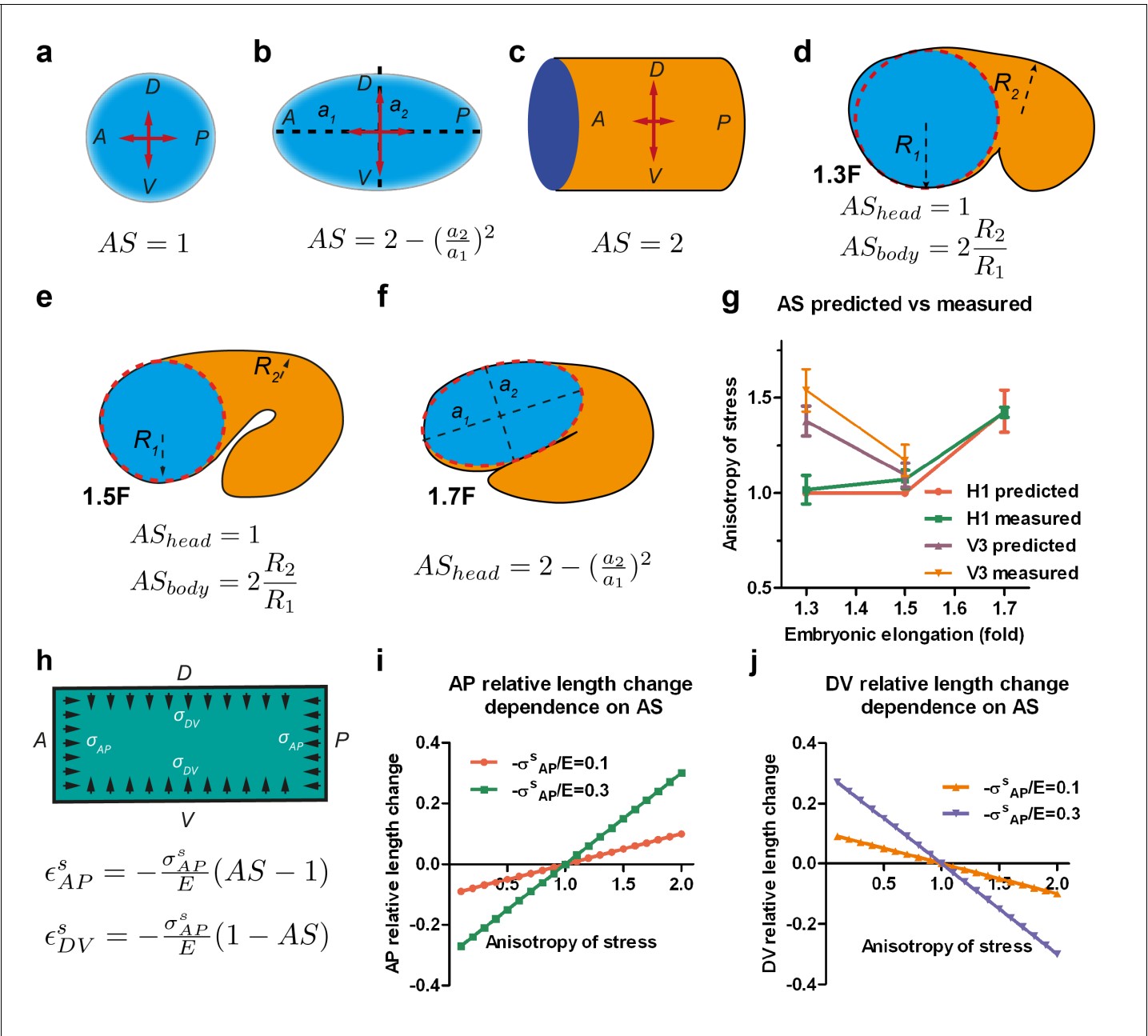

**Figure 4.** Stress anisotropy induces embryonic morphological changes. (a, b, c) Anisotropy of stress (AS) for a sphere (a), an ellipsoid (b) and a cylinder (c) with DV and AP axis defined in the schemes; (a, b) show the middle plane; the major and minor axis of the ellipsoid are called $a_1$ and $a_2$. (d, e, f) The embryo is schematized with a spherical (1.3F and 1.5F stages) or ellipsoidal (1.7F stage) head, and a curved cylindrical body. The AS in the head evolves from 1 (sphere) to that of an ellipsoid, whereas the body AS depends on the ratio of body to head radius ($R_2/R_1$). (g) Comparison of the predicted AS based on embryo diameter measurements (see *Figure 3c*) and the measured AS obtained from laser ablation experiments (*Figure 3b*). (h) Hooke's law written for an isotropic material like seam cells (Appendix 6A); $\epsilon_{AP}^s$ and $\epsilon_{DV}^s$ are the relative length changes along the AP and DV directions, respectively. The stress $\sigma_{AP}^s$ and $\sigma_{DV}^s$ along the AP and DV directions are supposed to be contractile (so negative). E, seam cell Young modulus; A, anterior; P, posterior; D, dorsal; V, ventral. (i, j) Dependence of the AP and DV relative length change on the AS for different values of $\sigma_{AP}^s/E$.

intermediate value between 1 and 2 for an ellipsoid. We can simplify the geometry of *C. elegans* embryos between the 1.3F and 1.5F stages as a curved cylinder (body), attached to a sphere (head) (*Figure 4d,e*). The head evolves into an ellipsoid between the 1.5F and 1.7F stages (*Figure 4f*). Thus, the AS of the head can be determined easily. We previously observed that the AP stress among the seam cells at a given stage differs by 20% (*Figure 3b*). Thus, if we approximate the AP stress as a constant at a given stage, the AS in the body will depend on the ratio of the body to head radius (*Figure 4d,e*, Appendix 5). Given the head and body diameter of the embryo (*Figure 3c*), we can compare the AS predicted by the thin-wall vessel model with those derived experimentally using laser ablation (*Figure 4g*). These values are nearly identical, showing that AS can be predicted on the basis of embryonic geometry.

To examine whether the AS can dictate embryonic morphological changes, we related the deformation of the vessel wall with the forces applied using the Hooke's law (*Figure 4h*, Appendix 6A) – for instance, Hooke's law states that the one-dimensional deformation of a spring is equal to the ratio of the applied force to the spring stiffness. Similarly, in a two-dimensional system and for an isotropic material, the deformation is proportional to the mechanical stress (forces) and inversely proportional to the Young modulus (stiffness) along the different loading directions (*Figure 4h*). The resulting equations, which assume that seam cells have an isotropic cortex and are subjected to contractile stress, correctly predict that the seam cell dimension increases along the AP axis ($\varepsilon_{AP}$) with the AS (*Figure 4h–j*), and decreases along the DV axis ($\varepsilon_{DV}$). Indeed, consistent with the equations, the head evolves from a sphere to an ellipsoid between the 1.5F to the 1.7F stages as the AS becomes greater than 1 (*Figure 3b,c*).

In conclusion, our experimental and modelling data show that the AS induces morphological changes in embryonic seam cells and provide a basis for understanding how the embryo elongates from a mechanical standpoint.

## Stiffness anisotropy-based elongation of the DV epidermis

As shown in *Figure 3e*, the head diameter reduction primarily involves changes in the circumferential width in the DV epidermis. As the RhoGAP RGA-2 maintains myosin II activation in these cells at a low level (*Diogon et al., 2007*), actomyosin contractility in DV cells cannot account for such changes. However, in contrast to seam cells, DV epidermal cells have circumferentially oriented actin bundles (*Figure 1b–d*), which based on recent observation could affect cell stiffness (*Calzado-Martín et al., 2016*; *Salker et al., 2016*). We thus hypothesized that the circumferential polarized actin distribution in DV epidermal cells could induce greater stiffness in that direction and thereby influences their deformation. To establish whether this is the case, we investigated both stress and stiffness distribution in the epidermal cells dorsal and ventral to the H1 seam cell using laser nano-ablation (*Figure 5a*). As these cells are the precursors of the HYP7 syncytium, we will denote them HYP7.

In the HYP7 cell, the opening in the DV direction was larger than that in the AP direction (*Figure 5b*; dorsal and ventral cells behaved similarly after laser cutting) at the 1.5F and 1.7F stages, as was the case in the H1 cell (*Figure 3b*). However, the ratio of DV/AP opening in HYP7 was greater than that in H1 (*Figure 5c*). Assuming that the HYP7 cell cortex has isotropic material properties like those of H1, our model (*Figure 4*; Appendix 5) would predict that the DV/AP opening ratio in HYP7 depends only on the head axisymmetric shape and is equal to that of H1, and would thus contradicts our observations. Hence, this suggests that the HYP7 cell has anisotropic cortical material properties.

To model the DV epidermal cell deformation, we examined two classes of anisotropic stiffness materials that have been described previously: orthotropic materials such as bones (*Miller et al., 2002*; *Helwig et al., 2009*), and fiber-reinforced materials such as arteries (*Gasser et al., 2006*), articular cartilage (*Federico and Gasser, 2010*) or fibrous connective tissues (*Ben Amar et al., 2015*). Orthotropic materials have different stiffnesses along orthogonal directions, and thus respond differently to the same stress magnitude along these directions. Fiber-reinforced materials also have different stiffnesses in the directions along and transverse to the fibers; in addition, such materials can respond differently to extensive or compressive stress (*Bert, 1977*). To define which model best applies to the DV epidermis, we used continuum linear elastic analysis (*Muskkhelishvili, 1975*; *Suo, 1990*; *Theocaris, 1986*; *Yoffe, 1951*) (Appendix 7) to interpret the laser-cutting data from the DV epidermis. We discarded the orthotropic model, as it did not

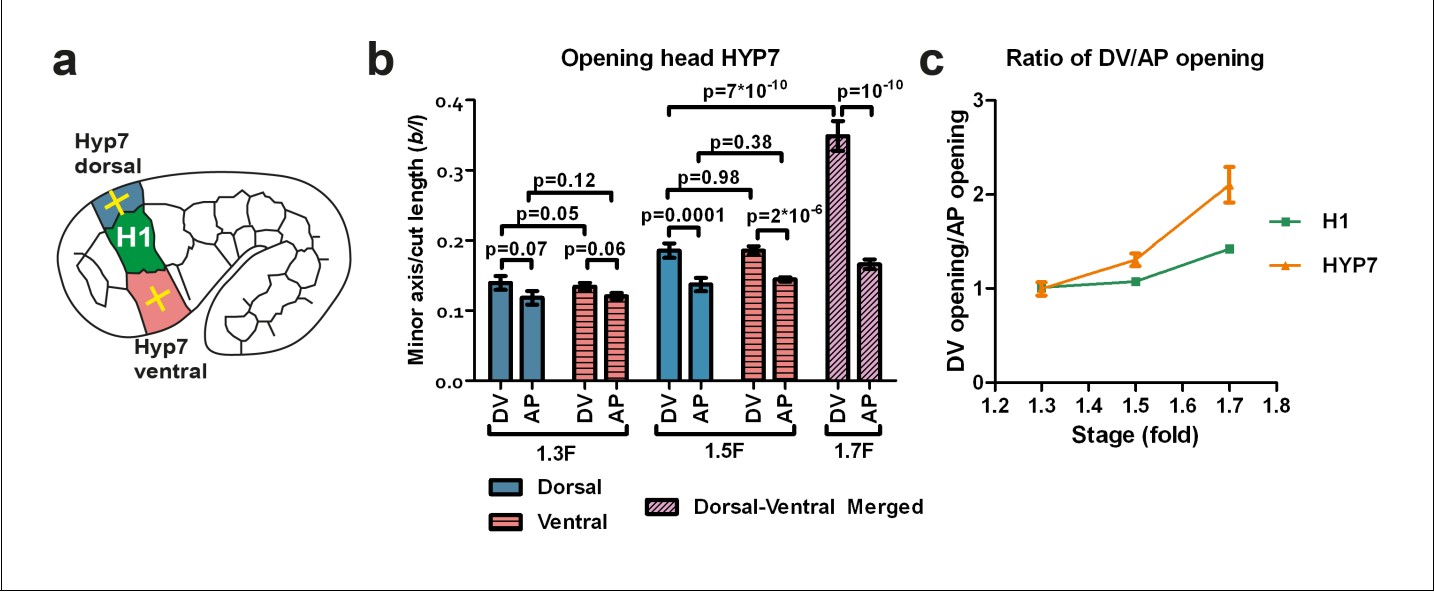

**Figure 5.** The dorso-ventral epidermis behaves differently than the H1 seam cell in ablation experiments. (**a**) Scheme showing laser ablation experiments in the epithelial cell HYP7 dorsal and ventral to H1, yellow crosses show cut directions. (**b**) Cut opening in the DV and AP directions measured in HYP7 between the 1.3F and 1.7F stages (see *Figure 2b*). p-values of two-tailed t-tests are reported. (**c**) Comparison of the DV/AP opening ratio in the seam cell H1 and in the head HYP7 cell. The data were derived from Figures 3b and 5b. To simplify, we will call the cells that will form the future HYP7 syncytium the HYP7 cells. The average (or computed) values and standard errors are reported. The numbers of embryos used for ablation are given in *Supplementary file 1*.

adequately describe our data (Appendix 8), and focused on the fiber-reinforced plane model, which better accounts for the presence of well aligned actin fibers in DV cells.

In a fiber-reinforced material that is composed of a matrix superimposed with fibers, the contribution of the fibers to the stiffness of the material depends on their orientation. Along the direction parallel to the fibers, the Young modulus of the composite is much increased due to fiber reinforcement, whereas along the direction perpendicular to the fibers, the contribution of the fibers to the composite stiffness is small. According to our modelling, the Young modulus along the fiber direction increases linearly with a factor $K$ related to the fiber stiffness and density; whereas the stiffness along the direction transverse to the fibers varies as a hyperbolic function of $K$ and reaches a plateau (Appendix 7). For fiber-reinforcement in the DV direction, the change in Young modulus along the DV and AP directions predicted by modelling is given in *Figure 6a,b*. Cuts perpendicular to the fibers opened similarly to an isotropic material with the matrix Young modulus, because they locally destroyed the fibers (*Figure 6c*; see *Equation (1)* above). By contrast, cuts along the fibers opened with an equilibrium value that depends on the fiber stiffness and distribution through the factor $K$ defined above (*Figure 6d*, Appendix 7).

Since the H1 seam and the head HYP7 cells are adjacent along the circumference (*Figure 3d*), they should be under the same DV stress due to tension continuity across cell-cell junctions. According to *Equation (1)*, if the stress in two cells is the same, their opening should vary inversely with their respective Young moduli. As the DV opening of HYP7 was about 1.5 times smaller than that of H1 (*Figure 7a*), we infer that the Young modulus of the HYP7 matrix without fibers was about 1.5 times stiffer than that of H1 (Appendix 9), suggesting that these cells have distinct material properties. Comparing the DV and AP opening for the HYP7 cell, we found that the factor $K$ increased during early elongation (*Figure 7b*; Appendix 10). More importantly, the calculated ratio of DV/AP Young moduli also increased, and was greater than the DV/AP stress anisotropy (*Figure 7c*).

To understand how a change in stiffness affects head HYP7 deformation, we again applied Hooke's law to these cells (Appendix 6B; *Figure 7d*). As myosin II activity in DV cells is low, their cortex should be exposed to tensile stress induced by actomyosin contractility in the seam cells. The cell length along the AP direction increased when the stiffness anisotropy (DV/AP stiffness

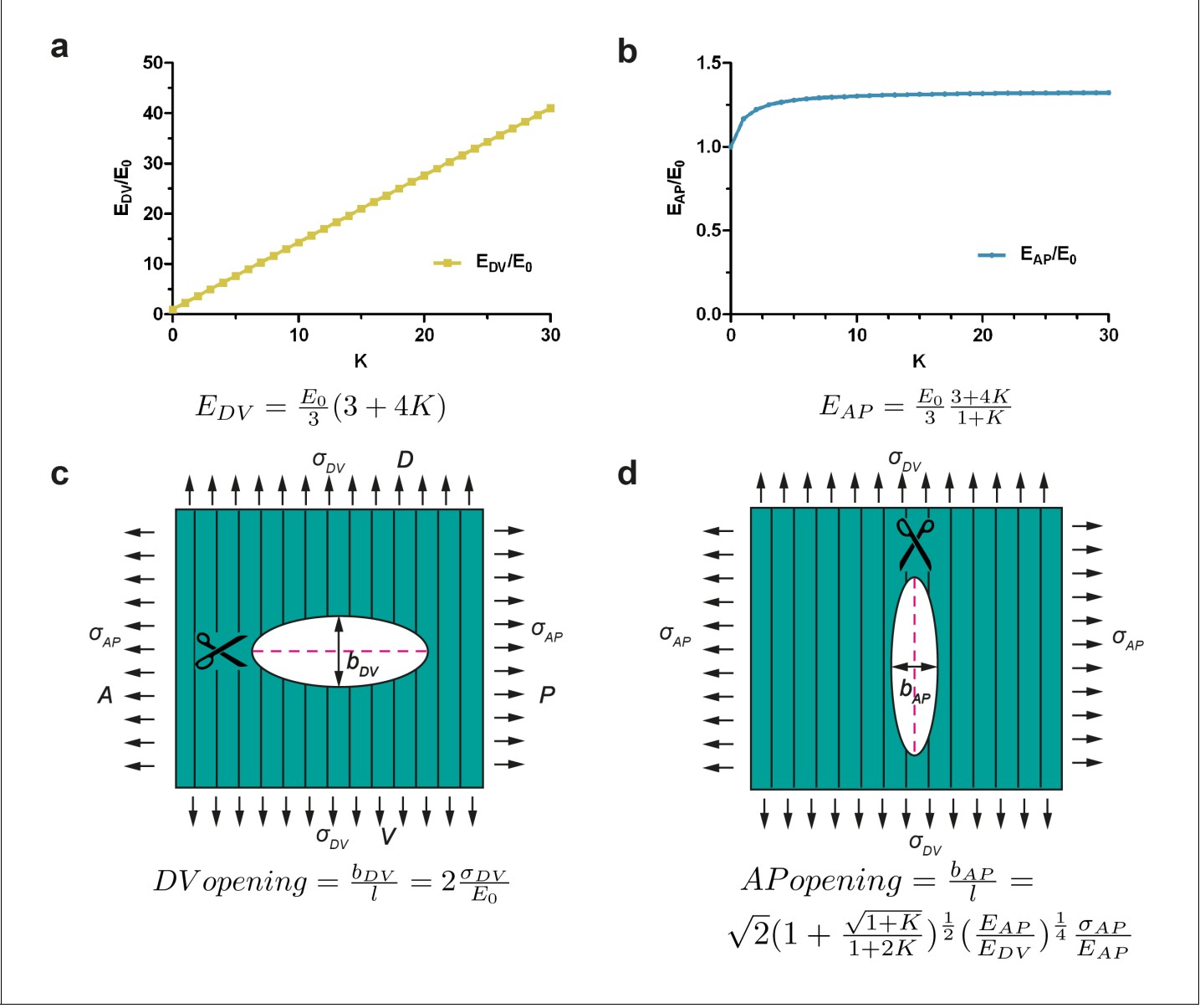

**Figure 6.** Model of cut opening for a fiber-reinforced material. (a) Considering a composite material with fiber reinforcement along the DV direction, the ratio of the Young modulus of the composite material along the DV direction, $E_{DV}$, to the Young modulus of the material without fibers (matrix), $E_0$, depends linearly on a factor $K$ (see Appendix 7); $K$ is related to fiber density and stiffness. (b) The ratio of the Young modulus along the AP direction, $E_{AP}$, to the matrix Young modulus, $E_0$, varies little with $K$. (c) The opening of the cuts perpendicular to the fibers is similar to an isotropic plane response, and depends on the ratio of DV stress to the matrix Young modulus $\sigma_{DV}/E_0$. (d) The opening of the cuts parallel to the fibers depends on the factor $K$, the ratio of AP/DV stiffness, $E_{AP}/E_{DV}$, and the ratio of stress over Young modulus in the AP direction, $\sigma_{AP}/E_{AP}$.

ratio) increased (*Figure 7d,e*), whereas the trend was opposite in the DV direction (*Figure 7d,f*). Thus, the stiffness anisotropy helps the HYP7 cell to extend along the AP direction and to shrink along the DV direction. Interestingly, the equations predict that increasing stress anisotropy has an opposite effect on HYP7 cell deformation, as it prevents these cells from extending antero-posteriorly (*Figure 7e*). Altogether, our model strongly suggests that when the DV/AP stiffness anisotropy increases and is greater than the DV/AP stress anisotropy, as observed in the head HYP7 (*Figure 7c*), elongation along the AP direction is favored. Furthermore, our data demonstrate that the distinct mechanical properties of the cells composing a complex tissue enables its morphogenesis , which does not require all cells to be contractile.

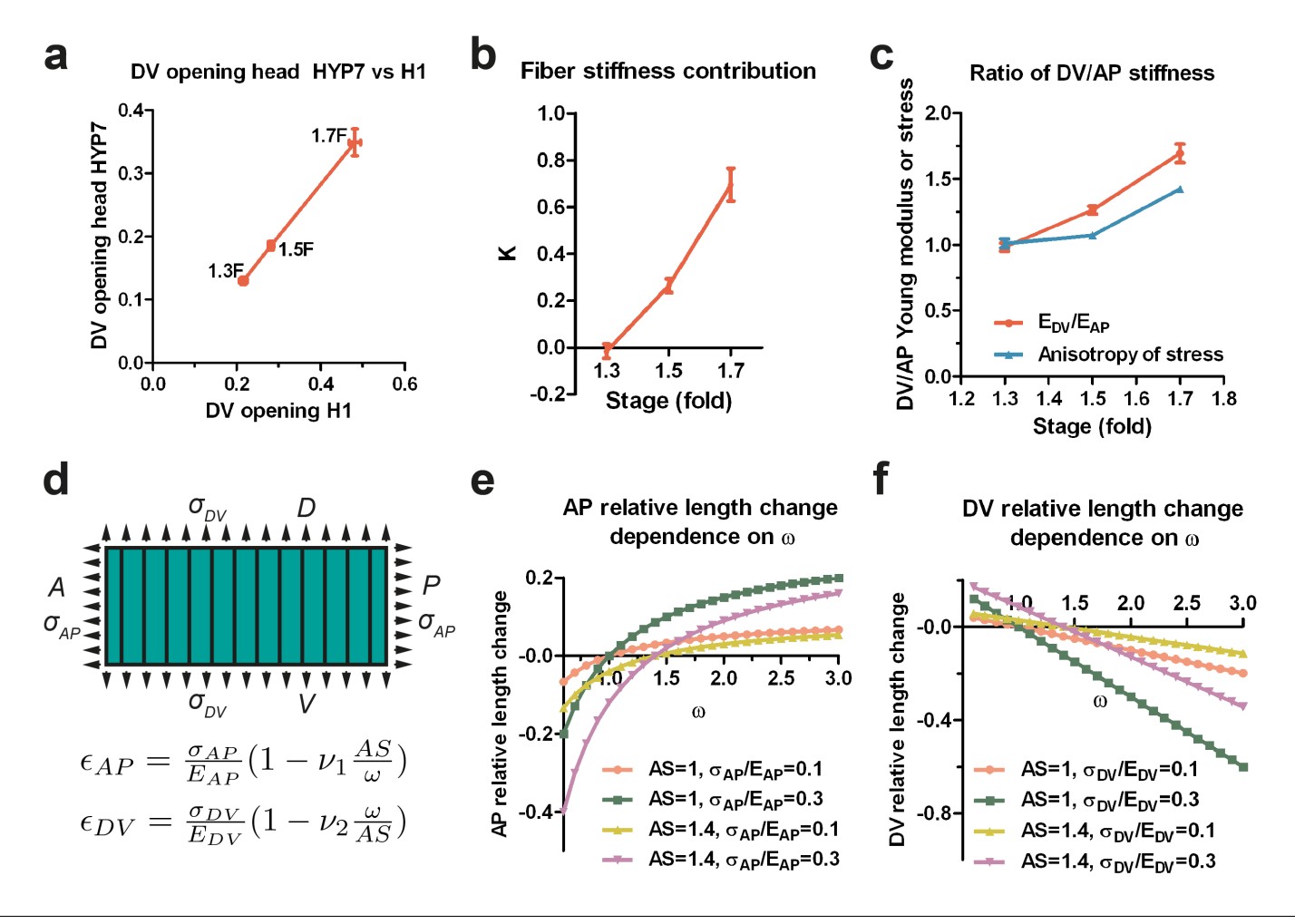

**Figure 7.** The anisotropy of stiffness in the HYP7 cell helps the embryo to elongate. (a) The cut opening in HYP7 is linearly related to the cut opening in H1. The slope of the linear regression gives the ratio of the HYP7 matrix without fibers to H1 Young moduli. (b) The factor $K$ increases during elongation from the 1.3F to 1.7F stages. (c) The ratio of DV/AP stiffness increases and is greater than the AS during early elongation. The data were derived from **Figure 3c** and Appendix 10. (d) Hooke's law written for a fiber-reinforced material such as DV cells (Appendix 6B). $\epsilon_{AP}$ and $\epsilon_{DV}$ are the relative length change along the AP and DV directions, respectively. The stresses $\sigma_{AP}$ and $\sigma_{DV}$ along the AP and DV directions, respectively are supposed to be tensile (so positive). $E_{AP}$ and $E_{DV}$ are the Young moduli in the DV cells along the AP and DV direction, respectively. $\nu_1$ and $\nu_2$ are Poisson's ratios in DV cells; $\omega$ is the $E_{DV}/E_{AP}$ ratio. A, anterior; P, posterior; D, dorsal; V, ventral. (e) Dependence of the AP relative length change $\epsilon_{AP}$ on the ratio of DV/AP stiffness $\omega$ for different values of AS and $\sigma_{AP}/E_{AP}$; the $\nu_1$ value is taken to be 1. (f) Dependence of the DV relative length change $\epsilon_{DV}$ on the ratio of DV/AP stiffness $\omega$ for different values of AS and $\sigma_{DV}/E_{DV}$; the $\nu_2$ value is taken to be 1. For (a–c), the average (or computed) values and standard errors are reported.

## Stress and stiffness anisotropies correlate with actin arrangement

Myosin II is not polarized (**Figure 1e,f**), but to find out whether actin distribution accounts for the stress and stiffness anisotropies, we carried out an analysis of actin filament alignment in the seam cells H1 and V3, as well as in the head HYP7 cell. We found that the polarization of actin filaments in seam cells correlated with the observed pattern of stress anisotropy. Indeed, in H1 at the 1.3F stage, actin filaments had a nearly isotropic angular distribution correlating with the isotropic stress (**Figure 8a**, **Figure 3b**), whereas they became increasingly aligned along the DV direction from the 1.3F to the 1.7F stage (**Figure 8a**), mirroring the increasing stress anisotropy from 1.3F to 1.7F (**Figure 3b**). Likewise, actin alignment decreased along the DV direction in V3 from the 1.3F to the 1.5F stage, in parallel to the decrease of stress anisotropy between those stages (**Figures 8b** and **3b**). The changes in H1 (**Figure 8c**), but not in V3 (**Figure 8d**), were statistically significant. In the

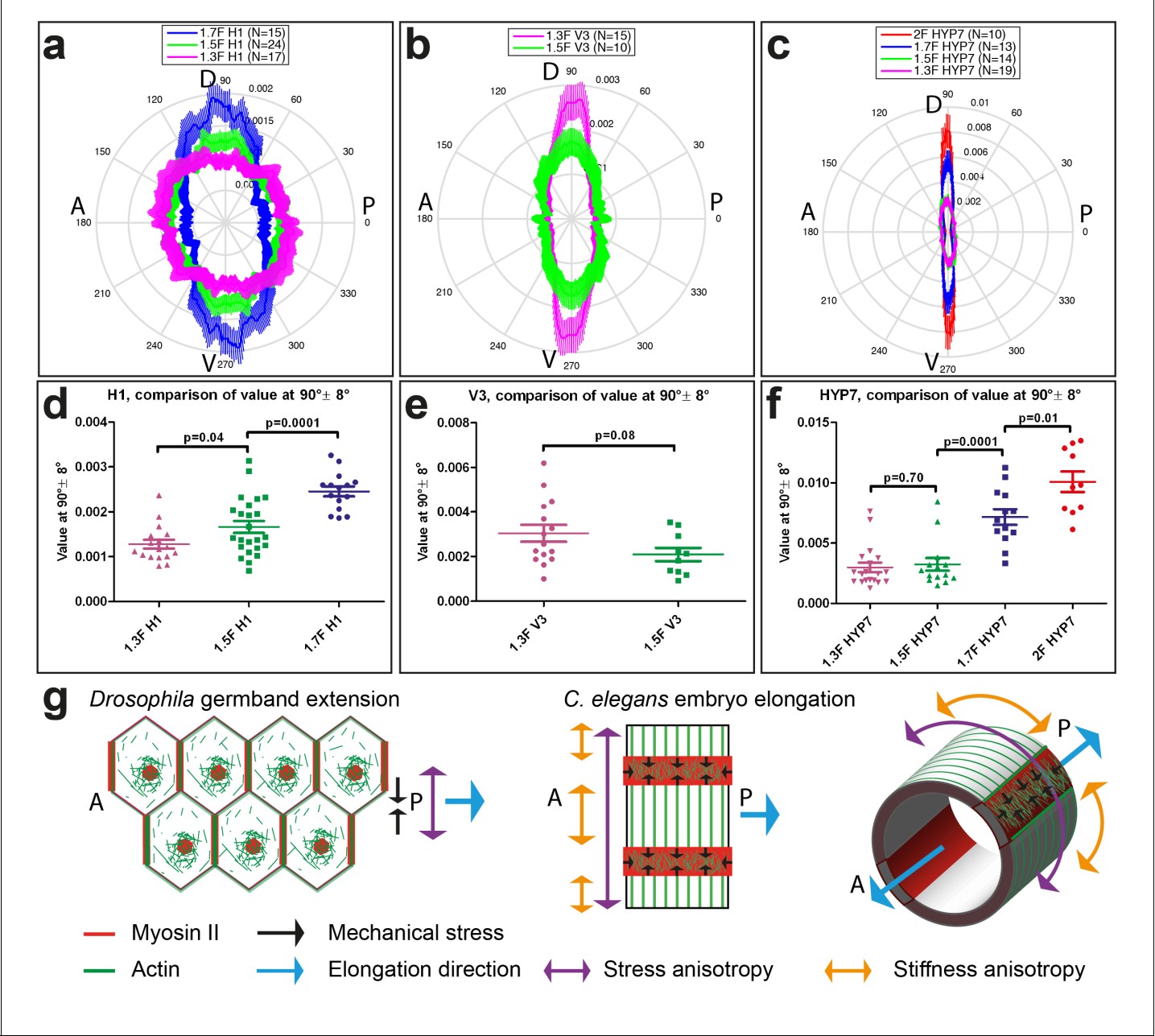

**Figure 8.** Actin filament organization correlated with stress and stiffness anisotropy pattern. (a–c) Angle distribution of actin filaments in the seam cell H1 (a), seam V3 (b) and the HYP7 cell (c), at different elongation stages. D, dorsal; V, ventral; A, anterior; P, posterior. 90° correspond to DV direction. (d, e, f) Comparison of the peak values at 90° ± 8° (DV direction) of angular distribution showed in (a, b, c), respectively. p-values of two-tailed t-tests are reported. (g) (Left) The anisotropy of mechanical stress generated by the polarized actomyosin network and medial myosin pulses promote *Drosophila* germband extension. (Right) The interplay of stress anisotropy (generated in seam cells - red) and stiffness anisotropy (DV cells – white) promote *C. elegans* embryo elongation. Note that, although myosin II does not display a polarized distribution within individual *C. elegans* epidermal cells as it does in *Drosophila* germband epithelial cells, its enrichment in seam cells along the circumference is reminiscent of the localized myosin II enrichment at vertical junctions in *Drosophila*. A, anterior; P, posterior. For (a–f), the average values and standard errors are reported.

HYP7 cell, actin filaments already acquired a preferential DV alignment at the 1.3F stage (*Figure 8c*), but became increasingly organized along the DV direction as the embryo elongated to the 2F stage, with a highly significant difference between the 1.5F and 1.7F stages (*Figure 8c and f*). These changes correlated with the increased stiffness anisotropy observed in the HYP7 cell (*Figure 7c*).

We have attempted to test functionally how actin organization could affect stress and stiffness by manipulating actin polymerization through two different strategies to express cofilin during early elongation. However, we could not obtain meaningful results. Altogether, we conclude that the pattern of actin distribution showed a good correlation with the observed stress and stiffness anisotropy. It will remain important to define the mechanisms that mediate changes in actin distribution, and ultimately to determine whether this distribution is a cause or a consequence of anisotropy.

## Discussion

Classical experiments in embryology have outlined how the juxtaposition of cells that have different properties is crucial in powering important morphogenetic movements, such as *Xenopus* gastrulation (*Hardin and Keller, 1988*; *Keller and Winklbauer, 1992*). In this work, we have dissected the mechanical contributions of the different epidermal cells driving *C. elegans* embryonic morphogenesis at single-cell resolution, highlighting the importance of juxtaposition of cells with different properties. Combining laser nano-ablation and continuum mechanics modelling, we first highlight the importance of stress anisotropy in the seam cells. Second, we emphasize that stiffness anisotropy is equally important for embryonic elongation but matters in another epidermal cell type, the DV cells. Thereby, we reveal the critical role of tissue material properties in morphogenesis.

Many studies analysing morphogenetic processes have focused on 2D epithelial sheets such as the *Drosophila* mesoderm (*Martin et al., 2009*), germband (*Rauzi et al., 2008*, *2010*; *Blankenship et al., 2006*; *Fernandez-Gonzalez and Zallen, 2011*), amnioserosa (*Solon et al., 2009*; *Gorfinkiel et al., 2009*), wing and thorax (*Aigouy et al., 2010*; *Bosveld et al., 2012*), or the zebrafish enveloping cell layer (*Behrndt et al., 2012*) during embryonic development. They have revealed the role of contractile actomyosin pulses and planar polarity in coordinating events over long distances. The *C. elegans* embryonic elongation is distinct from those situations because it does not involve myosin-II-polarized distribution nor actomyosin pulses. Interestingly, this process still requires stress anisotropy, outlining that stress anisotropy can be produced by different means.

We suggest that several factors contribute to establish stress anisotropy in *C. elegans*. First, the actin network displayed a more polarized dorso-ventral distribution in seam cells when the stress anisotropy was higher, which should increase the stress in that direction. Second, akin to a planar polarized distribution, myosin II activity displays an asymmetric distribution along the embryo circumference in cells with different material properties (*Figure 8g*). Intriguingly, tissue culture cells can sense the spatial stiffness distribution (*Walcott and Sun, 2010*; *Fouchard et al., 2011*; *Trichet et al., 2012*; *Lange and Fabry, 2013*), raising the possibility that seam cells sense the higher circumferential stiffness and respond with higher DV-oriented stress through the mechano-sensitive adherens junctions (*le Duc et al., 2010*; *Yonemura et al., 2010*). Third, we found that the spectrin cytoskeleton has a significant role in establishing normal levels of stress magnitude and anisotropy. Spectrin is known to impinge on actin filament alignment and continuity in DV cells (*Praitis et al., 2005*; *Norman and Moerman, 2002*) and could thus affect DV stiffness anisotropy by reducing the level of actin fiber alignment. Finally, although myosin II activity is low in DV cells (*Diogon et al., 2007*), the remaining activity might create some DV-oriented stress feeding back on seam cells.

By modelling the DV cells as a fiber-reinforced material, we reveal how the polarized cytoskeleton in DV cells increases their stiffness to orient the extension in the AP direction, acting like a 'molecular corset'. Related 'molecular corsets' have been described and proposed to drive axis elongation in other systems (*Wainwright, 1988*). In *Drosophila*, a network of extracellular matrix fibrils was proposed to help elongate developing eggs (*Haigo and Bilder, 2011*). In plant cells, the orientation of cellulose microfibrils determines the axis of maximal expansion. In the latter, stiffness anisotropy also helps overcome stress anisotropy (*Green, 1962*; *Baskin, 2005*). Importantly, *C. elegans* embryos reduce their circumference during elongation, whereas *Drosophila* eggs and plants increase it. This suggests that to conserve the actin reinforcement properties when the diameter decreases, *C. elegans* DV epidermal cells should have a mechanism to actively shorten the actin bundles, as observed in a biomimetic in vitro system (*Murrell and Gardel, 2012*).

Our experimental data were consistent with the predictions from Hooke's law. They prove that the actomyosin cortex preferentially squeezes the embryo circumferentially, and that the stress anisotropy is tightly linked to the geometry of the embryo. By quantitatively assessing the

contribution of stiffness anisotropy in tissue elongation, we have emphasized its importance relative to the more established role of stress anisotropy. The precise relationship between both anisotropies remains to be investigated. Thus, the juxtaposition of cells with different 'physical phenotypes', seam epidermis expressing stress anisotropy and DV epidermal cell showing stiffness anisotropy, powers *C. elegans* elongation, as previously suggested in chicken limb bud outgrowth (*Damon et al., 2008*) or chick intestinal looping (*Savin et al., 2011*). We did not mention other potential stress-bearing components, such as microtubules and the embryonic sheath (*Priess and Hirsh, 1986*), as the former mainly serves to enable protein transport (*Quintin et al., 2016*) whereas the function of the latter will be the focus of an upcoming work.

In conclusion, our work shows that tissue elongation relies on two fundamental physical quantities (mechanical stress and tissue stiffness), and provides the most advanced mesoscopic understanding to date of the mechanics at work during the first steps of *C. elegans* embryonic elongation.

## Materials and methods

### *C. elegans* alleles and strains

Bristol N2 was used as the wild-type (WT) strain and animals were maintained as described in *Brenner (1974)*. The strain ML1540: *mcIs50[lin-26p::vab-10(abd)::gfp; myo-2p::gfp]* LGI carrying the actin-binding domain (ABD) of the protein VAB-10 under the epidermal promoter *lin-26* has been described elsewhere (*Gally et al., 2009*). The endogenous NMY-1::GFP reporter strain was built by CRISPR knock-in (ML2540: *nmy-1(mc82)[nmy-1::gfp]* LGX; the NMY-2::GFP reporter strain LP162 *nmy-2(cp13)[nmy-2::gfp + LoxP]* LGI was a generous gift from Daniel Dickinson.

For parallel calcium and actin imaging during ablation, we used the strain ML2142: *mcIs43 [lin26p:: vab-10::mCherry; myo-2p::gfp]; juIs307[dpy-7p::GCaMP3]* carrying a calcium sensor under the epidermal promoter *dpy-7p* and mCherry-labeled VAB-10(ABD) under the *lin-26* promoter. The thermosensitive Rho kinase mutation *let-502(sb118ts)* was crossed with ML1540 to give the strain ML2216: *let-502(sb118ts); mcIs50[lin-26p::vab-10(abd)::gfp; myo-2p::gfp]* LGI.

For determining morphological changes, we used the strain ML2386: *mcIs50[lin-26p::vab-10 (abd)::gfp; myo-2p::gfp] I; xnIs97[hmr-1::gfp] III*, which expresses both a junctional marker (HMR-1/E-cadherin) and an actin marker (VAB-10(ABD)).

For actin alignment analysis, we used the ML1966 *unc-119(ed3) mcIs67 [dpy7p::LifeAct::GFP; unc-119(+)]* strain, expressing the actin reporter LIFEACT under the *dpy-7* epidermal promoter.

### RNA interference

RNAi experiments were done using injection of double-stranded RNA synthesized from PCR-amplified genomic fragments using a T3 or T7 mMESSAGE mMACHINE Kit (Ambion, Austin, TX, USA). The embryos were analyzed from 24 hr to 48 hr post-injection.

### Time-lapse analysis and morphological change quantification

Freshly laid embryos or embryos from dissected hermaphrodites were mounted on 5% agarose pads in M9 buffer and the coverslip was sealed with paraffin oil. DIC time-lapse movies were recorded at 20°C using a Leica DM6000 upright microscope with a 40X oil immersion objective. For each embryo, a Z-stack of 7–8 focal planes with 4 µm step size was acquired. The length of embryos was estimated by tracing the embryo body axis (through the middle of the embryo). Fluorescence time-lapse movies were recorded at 20°C using a spinning-disk Zeiss microscope Axio Observer.Z1 using a 63X oil immersion objective. Other fluorescence images were acquired with the same microscope using a 100X oil immersion objective. To determine the morphological changes in the embryo, sections of the embryo imaged with junctional and actin markers at the level of H1, V3 and V6 were reconstructed to determine the radius, seam and DV cell width along the circumferential direction. All images were analyzed using the ImageJ (FiJi) software (NIH, Bethesda, Maryland, USA; http://rsb.info.nih.gov/ij/) and MATLAB R2014b (The MathWorks Inc., Natick, MA).

### Actin alignment analysis

Z-stack images of LIFEACT::GFP fluorescence expression in the epidermis were acquired using a confocal Leica SP5 microscope with a 63X oil immersion objective and zoom factor 8. We used a

step size of 0.08 µm, a pinhole opening of 0.6 Airy Unit and projected 2 µm around the actin cortex. The embryos were rotated on the scan field to have the same antero-posterior orientation. The acquired images were deconvoluted using the Huygens Essential software from Scientific Volume Imaging (Hilversum, Netherlands). We chose a region of interest (ROI) of $4 \times 4$ µm$^2$ within the seam cell H1 or dorso-ventral epidermal cell HYP7, and of $3 \times 3$ µm$^2$ within the seam cell V3 to perform Fast Fourier Transform (FFT). We used a high-pass filter to remove the low frequencies then did inverse FFT. We found that the high pass filter removed changes in intensity due to unequal labelling or out of focus signals but retained the actin texture. Finally, we used an ImageJ plugin, 'Spectral Texture Analysis', written by Julien Pontabry to derive the angle distribution of actin texture. This plugin performed the Fast Fourier Transform (FFT) of the given ROI and computing coefficients in Fourier space, such as the angle distribution of the given structure, as detailed in *Gonzalez and Woods (2008)*. Digital image processing, Nueva Jersey, chapter 11, section 3.3 (*Gonzalez, 2008*).

## Embryo staging for ablation

For ablations, we compared embryos of the same developmental timing. To do so, we recorded the elongation curve of different genetics background (*Figure 3—figure supplement 1*) and took embryos at the corresponding developmental time from the beginning of elongation. Thus, *unc-112 (RNAi)* embryos elongating up to 1.7F similarly to WT have the same length as WT at 1.7F stage. By contrast, *spc-1(RNAi)* embryos elongated slower than WT (*Figure 3—figure supplement 1*) and thus at a time corresponding to 1.7F in a control embryos were shorter than wild-type embryos at 1.7F stage. For comparison between *let-502(sb118ts)* embryos, measurements were carried out at 25.5°C and the embryos were taken when muscles started to twitch (at around 1.5F in control embryos).

## Laser ablation

Laser ablation was performed using a Leica TCS SP8 Confocal Laser Scanning microscope, with a femtosecond near-infrared Coherent Chameleon Vision II, Ti:Sapphire 680–1080 nm laser, 80 MHz. To make a line cut, a region of interest with a length varying from 3 µm to 6 µm and a width of 0.08 µm (1 pixel width) was drawn. We used a laser wavelength varying from 800 nm to 900 nm, which gave consistent ablation responses. The laser power was tuned before each imaging section to obtain local disruption of the cortex response (>80% of the cases, visible opening, no actin accumulation around cell borders in the repair process and the ablated embryos developed normally). Typically, the power of the laser was 2000 mW, and we used 50% power at 100% gain. Wounding response (actin accumulation around cell borders in the repair process, embryo died afterwards) was rarely observed at the power used for local disruption, but more often when the power was increased to 60–65%. The first time point was recorded 1.44 s after cutting, which corresponded to the time needed to reset the microscope from a two-photon to a regular imaging configuration. The image scanning time recorded by the software was usually less than 400 ms, so the total exposure time of the chosen ROI to multiphoton laser was less than 1 ms. The cuts were oriented either in the antero-posterior (AP) or dorso-ventral (DV) directions relative to the global orientation of the embryos. After ablation, the embryos were monitored to see whether they continued to develop normally or whether they expressed the desired phenotype. More precisely, we verified whether embryos ablated at 1.3F and 1.5F developed past the 2F stage, whether embryos ablated at 1.7F developed past the 2.5F stage, and whether *unc-112(RNAi)* embryos and *spc-1(RNAi)* embryos arrested at 2F stage.

## Laser ablation image processing and data analysis

The shape of the cut opening was detected using the Active Contour plugin ABsnake (*Boudier, 1997*). A starting ROI was drawn around the opening as the initiation ROI for ABsnake. After running the plugin, the results were checked and corrected for detection errors. The detected shape was fitted with an ellipse to derive the minor axis, major axis and the angle formed by the major axis with the initial cut direction. The average opening of the five last time points before the repair process began (*Figure 2—figure supplement 1*, from around 8 to 10 s after cutting) was taken as the opening at equilibrium. The standard error of the mean is reported.

The curve fit was performed on the average value of the cut opening (defined as the minor axis/ initial cut length) using GraphPad Prism 5.00 (San Diego, California, USA) and the equation of one-phase association:

$$y = y_0 + (Plateau - y_0) * (1 - e^{-\gamma t})$$

where $y_0$ is the initial width of the cut opening, *Plateau* is the minor axis of the opening at equilibrium and $\gamma$ is the relaxation rate. The standard error of the mean given by the software is reported.

## Statistical analysis

The two-tailed t-test was performed on the average of the last five time points (from about 8 s to 10 s) of the cut opening using MATLAB R2014b (The MathWorks Inc., Natick, MA). Z-test was performed using QuickCalcs of GraphPad Prism (San Diego, California, USA) to compare the anisotropy of stress (AS), the relaxation half-time and the initial recoil speed of the cut opening.

## Acknowledgements

The authors thank the Imaging Service of IGBMC, Demet Kirmizibayrak and Marcel Boeglin for technical assistance; Pierre-François Lenne for help in laser ablation setup and reading the manuscript; and Flora Llense, Teresa Ferraro, François Robin, Sylvie Schneider-Maunoury and Raphaël Voituriez for critical comments on the manuscript.

This work was supported by an European Research Council grant to ML (grant #294744), and by institutional funds from the Centre National de la Recherche Scientifique (CNRS), University of Strasbourg and University Pierre et Marie Curie (UPMC), by the grant ANR-10-LABX-0030-INRT, which is a French State fund managed by the Agence Nationale de la Recherche under the framework programme Investissements d'Avenir labelled ANR-10-IDEX-0002–02 to the IGBMC, and by installation grants from the CNRS and UPMC to ML. MBA is supported in part by Institut Universitaire de France. Some strains were obtained from the *Caenorhabditis* Genetics Center CGC (funded by the NIH Office of Research Infrastructure Programs P40 OD010440). Some confocal work was carried out at the Institute of Biology Paris Seine Imaging facility, which is significantly supported by the 'Conseil Regional Ile-de-France', the French national research council (CNRS) and Sorbonne University, UPMC Univ Paris 06.

## Additional information

### Funding

| Funder | Grant reference number | Author |
| --- | --- | --- |
| European Research Council | #294744 | Michel Labouesse |
| Centre National de la Recherche Scientifique | ANR-10-LABX-0030-INRT | Michel Labouesse |
| Université de Strasbourg | ANR-10-IDEX-0002-02 | Michel Labouesse |
| Université Pierre et Marie Curie | ANR-10-LABX-0030-INRT | Michel Labouesse |

The funders had no role in study design, data collection and interpretation, or the decision to submit the work for publication.

### Author contributions

TTKV-B, Conceptualization, Resources, Data curation, Formal analysis, Validation, Investigation, Visualization, Methodology, Writing—original draft, Writing—review and editing, Modelling; MBA, Validation, Methodology, Writing—original draft, Modelling; JP, Software; ML, Conceptualization, Resources, Data curation, Supervision, Funding acquisition, Validation, Investigation, Methodology, Writing—original draft, Project administration, Writing—review and editing

**Author ORCIDs**

Thanh Thi Kim Vuong-Brender, http://orcid.org/0000-0001-6594-2881

Martine Ben Amar, http://orcid.org/0000-0001-9132-2053

Julien Pontabry, http://orcid.org/0000-0001-7412-4645

Michel Labouesse, http://orcid.org/0000-0001-7995-5843

## Additional files

**Supplementary files**

• Supplementary file 1. Number of embryos used for laser ablation.

• Supplementary file 2. Goodness of fit ($R^2$) of cut border relaxation with different initial width of the cut openings.

• Supplementary file 3. Parameters derived from single exponential fit of the cut relaxation with an initial width of cut openings of 0.6 µm. (see Materials and methods).

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

## Appendix 1

# An overview of the early phase of *C. elegans* embryonic elongation

Soon after the ventral enclosure has completed, *C. elegans* embryos elongate from a lima-bean shape to the characteristic cylindrical shape, resulting in a 4-fold increase in length and approximately a 2-fold decrease in diameter (*Priess and Hirsh, 1986*) (*Figure 1a*). The elongation is thought to be driven by cell shape changes, as can be most easily observed among seam cells (*Figure 1a*). Muscle contractions, starting about midway through the process, are essential: muscle-defective embryos are paralyzed and arrest at the 2F stage. In the following paragraphs, we mostly discuss the early phase of elongation that occurs prior to the onset of muscle contraction.

Pharmacological and genetic studies have proved the critical role of actomyosin contractility during the early elongation. Inhibition of actin polymerization with cytochalasin-D prior to the 1.5F stage blocks elongation, whereas application at later stage causes embryos to retract to their pre-elongation state (*Priess and Hirsh, 1986*). Non-muscle myosin II activation is regulated through phosphorylation and dephosphorylation of the regulatory light chain MLC-4 by the LET-502/Rho-binding kinase and the MEL-11/Myosin phosphatase, respectively (*Wissmann et al., 1997*, *1999*; *Gally et al., 2009*). LET-502, the effector of the Rho GTPase RHO-1, can be activated by the *C. elegans* RhoGEF (Guanine Exchange Factors) RHGF-2 and inactivated by the RhoGAP (GTPase Activating Protein) RGA-2 (*Chan et al., 2015*; *Diogon et al., 2007*). Mutations affecting myosin II or its activation, such as MLC-4 and NMY-1/Non-muscle Myosin II heavy chain, RHGF-2/RhoGEF, and LET-502/ROCK, lead to hypo-elongation, causing embryos to arrest earlier than or at the 2F stage (*Wissmann et al., 1997*; *Gally et al., 2009*; *Chan et al., 2015*; *Shelton et al., 1999*; *Piekny et al., 2003*). By contrast, mutations affecting negative regulators of myosin II, such as *mel-11* or *rga-2*, cause embryos to burst during elongation due to increased tension exerted on adherens junctions (*Wissmann et al., 1999*; *Diogon et al., 2007*). The two myosin heavy chains, NMY-1 and NMY-2, work redundantly to regulate actomyosin contractility (*Piekny et al., 2003*). Although essential during early embryonic development, our data show that NMY-2 is not required during elongation. Indeed, embryos that are homozygous for the strong thermosensitive mutant *nmy-2*(*ne3409*) (*Liu et al., 2010*) still elongated normally after shifting to the restrictive temperature (25.5° C data not shown).

Several lines of evidence suggest that seam cells generate most of the actomyosin forces, while the DV cells may remain passive. First, the myosin II regulatory light chain MLC-4 is mainly required in seam cells (*Gally et al., 2009*). Second, MLC-4, MLC-5/myosin essential light chain and NMY-1 are expressed at higher levels in seam cells (*Gally et al., 2009*; *Shelton et al., 1999*; *Piekny et al., 2003*). Third, rescue experiments have shown that the positive regulator of contractility RHGF-2/RhoGEF is required only in seam cells, whereas the negative regulator RGA-2/RhoGAP acts specifically in DV cells (*Shelton et al., 1999*; *Piekny et al., 2003*). Thus, all the players of acto-myosin regulation pathway promote a high contractility in seam cells and keep a low contractility in DV cells.

Although the myosin II activity is crucial in seam cells, the phenotypes of several mutants affecting junctional proteins, which are thought to anchor actin bundles (*Costa et al., 1998*), show an important role of actin bundles in DV epidermal cells during elongation. In particular, zygotic *hmp-1/α-catenin* mutants, in which actin bundles detach from the junctional belt, show bulges and cannot elongate. Similarly, loss of ZOO-1/ZO-1 and VAB-9/Claudin homologues affects actin bundle organization in DV cells, leading to deformities and an incomplete elongation (*Lockwood et al., 2008*; *Simske et al., 2003*).

In summary, the epidermal actomyosin network is essential for the early elongation phase of the *C. elegans* embryo.

**Appendix 2**

## The *C. elegans* embryonic epidermis is under biaxial stress loading along the AP and DV directions

Biological materials are in general viscoelastic (*Kasza et al., 2007*), but the elastic aspect of *C. elegans* embryos seems to be more important. Indeed, inhibition of actin polymerization with cytochalasin-D induces a retraction of the embryo to nearly its original length (*Priess and Hirsh, 1986*), like a spring after force release. Thus, we used an elastic model to describe *C. elegans* embryonic deformation. In particular, we considered the epidermal cell cortex as an elastic plane. In this section, we examine, for the analysis of laser ablation responses, whether the epidermal cell cortex is subjected to biaxial stress loading (stress along two orthogonal directions) along the AP and DV directions.

If a thin cut is introduced in an infinite isotropic elastic plane under biaxial stress loading (*Appendix 2—figure1a*), Theocaris et al. (*1986*) have shown that the opening is an ellipse. These authors have shown that the rotation angle $\theta$ between the direction of the cut and the major axis of the opening ellipse is in general different from zero (*Appendix 2—figure1b*). $\theta$ is equal to zero when the cuts are parallel to the directions of stress loading or when there is equal tension-tension loading ($\sigma_{xx} = \sigma_{yy} > 0$) (*Theocaris et al., 1986*).

To test whether the AP and DV directions are indeed the directions of stress loading for different seam cells (H1, V3 and V6; *Figure 3a*) during early elongation, we performed laser cuts in the AP and DV directions and measured the rotation angle $\theta$ at equilibrium from the 1.3F to the 1.7F stages. *Appendix 2—figure 1c* shows that $\theta$ was not significantly different from zero for different magnitudes of stress (*Figure 3b*), consistent with the hypothesis that AP and DV are the principal directions of stress loading. The difference of $\theta$ compared to 0 was more important for V6, as we had difficulties in determining unambiguously the AP and DV direction for V6. In conclusion, the epidermal cell cortex can be considered as an elastic plane under biaxial stress loading along the AP and DV directions.

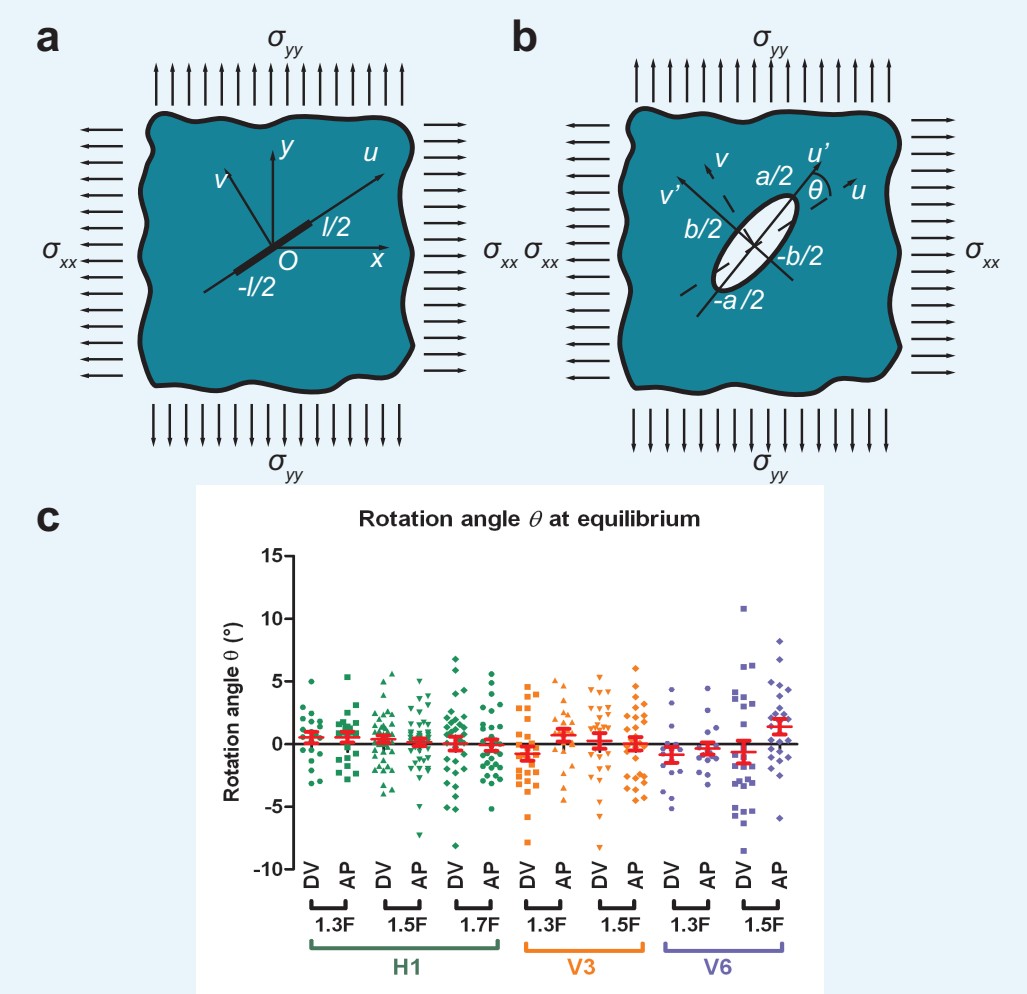

**Appendix 2—figure 1.** Determination of stress loading directions from cut shape rotation.
(**a**) An incision of length $l$ was introduced in an elastic plane under biaxial stress along the $x$
and $y$ directions. The cut was along the $u$ axis with the $uv$ coordinates. (**b**) The shape of the
opening at equilibrium was an ellipse (*Theocaris et al., 1986*). We called $u'$ (of $u'v'$
coordinates) the axis parallel to the major axis of the opening ellipse, which formed an angle
$\theta$ to the direction of the initial cut. (**c**) The rotation angle $\theta$ at equilibrium (measured around
9.5 s after cut) for H1 from the 1.3F to the 1.7F stages and for V3 and V6 from the 1.3F to
the 1.5F stages. 'DV' and 'AP' mean DV and AP opening, respectively. Red bars show the
means and standard errors.

## Appendix 3

# Comparison of two methods to analyze laser ablation responses using the recoil dynamics and the cut opening at equilibrium

In order to compare the two analysis methods of laser ablation using the equilibrium shape of the cut opening or the recoil dynamics (**Rauzi and Lenne, 2015**; **Smutny et al., 2015**), we estimated the initial recoil speed and the relaxation half-time by fitting the relaxation of the cut borders (the minor axis of the cut opening) with the equation:

$$y = y_0 + (Plateau - y_0)(1 - e^{-\gamma t}) \tag{1}$$

where $y_0$ is the initial width of the cut opening, *Plateau* is the minor axis of the cut opening at equilibrium and $\gamma$ is the relaxation rate. The recoil speed can be obtained by taking the derivation of the previous equation versus time:

$$v = \frac{dy}{dt} = (Plateau - y_0)\gamma e^{-\gamma t} \tag{2}$$

Thus, the initial recoil speed is:

$$v_0 = (Plateau - y_0)\gamma \tag{3}$$

The half-time, defined as the time interval needed to reach half of the distance between the initial opening and the *Plateau*, is given by:

$$\tau_{\frac{1}{2}} = \frac{ln(2)}{\gamma} \tag{4}$$

According to the model described in **Rauzi and Lenne (2015)**, **Smutny et al. (2015)** and **Mayer et al. (2010)**, the initial recoil speed is proportional to the ratio of the cortical stress $\sigma$ to the viscosity of the cellular medium $\eta$:

$$v_0 \sim \frac{\sigma}{\eta} \tag{5}$$

and the relaxation half-time depends on the ratio of $\eta$ to the cortex stiffness $k$:

$$\tau_{\frac{1}{2}} \sim \frac{\eta}{k} \tag{6}$$

The fitting of the **Equation (1)** depends on the initial width of the cut opening $y_0$. We used three methods to estimate $y_0$. First, since the cut opening depends on the actomyosin contractility (**Figure 2c**), we reasoned that $y_0$ should be close to the opening observed in a mutant where actomyosin contractility is strongly inhibited. Indeed, in the *let-502(sb118ts)* mutant, the opening changed very little (**Figure 2c**), which corresponded to a nearly complete inhibition of the early elongation. The smallest opening in the *let-502(sb118ts)* mutant was around 0.6 $\mu m$ (**Figure 2c**). Second, our evaluation of the initial width of the opening using the same setup to photobleach a thin fluorescent layer also gave an opening of around 0.6 $\mu m$. Finally, we tried to fit the recoil of cut borders using **Equation (1)** with values of $y_0$ decreasing from 0.6 $\mu m$ to 0.2 $\mu m$ (in 0.1 $\mu m$ steps) and found a decrease of

goodness of fit $R^2$ (Supplementary table 2). The goodness of fit $R^2$ is a fraction between 0 and 1, and higher values indicate better fits (GraphPad Prism 5, *Goodness of fit of nonlinear regression*). Thus, all three approaches indicated that 0.6 $\mu m$ was a good estimation of the initial width of the cut opening (an example of fit is shown in **Figure 2—figure supplement 1**). Subsequently, we used $y_0 = 0.6$ $\mu$m to derive the initial recoil speed and the relaxation half-time, and the results are shown in Supplementary table 3.

We found that the relaxation half-time in the AP direction was similar with that in the DV direction in most of the seam cells examined at different stages (**Appendix 3—figure 1a**). As the relaxation time is proportional to the ratio of viscosity over the stiffness of the cortex, and the cytoplasmic viscosity is likely to be homogeneous within a given cell, the cortex in seam cells is likely to be isotropic. However, it can vary from one cell to another as indicated in **Appendix 3—figure 1a**.

Next, we wanted to know whether the two methods to analyze the ablation response (based on the cut recoil dynamics and the cut opening at equilibrium) gave consistent results for the stress magnitude and anisotropy. As we compared different seam cells (H1, V3 and V6) with potentially different material properties (viscosity and stiffness), we multiplied the initial recoil speed by the relaxation half-time to have the stress to stiffness ratio (**Equations [5, 6]**). We normalized the previous ratio to the different cut lengths used in the different seam cells (5 $\mu m$ in H1, 4 $\mu m$ in V3 and V6), then plotted the resulting values against the cut opening at equilibrium, which also reports the stress over stiffness ratio (**Figure 2b**, **Appendix 3—figure 1a,b,d**). The linear regression strongly suggests that the two methods are in good agreement on the stress magnitude. Moreover, the anisotropy of stress obtained by the two methods showed a linear correlation (**Appendix 3—figure 1e**).

In summary, the two methods to analyze laser ablation responses gave results consistent with each other on the magnitude and the anisotropy of stress. The relaxation half-time obtained from recoil dynamics analysis indicated that the seam cell cortex is isotropic.

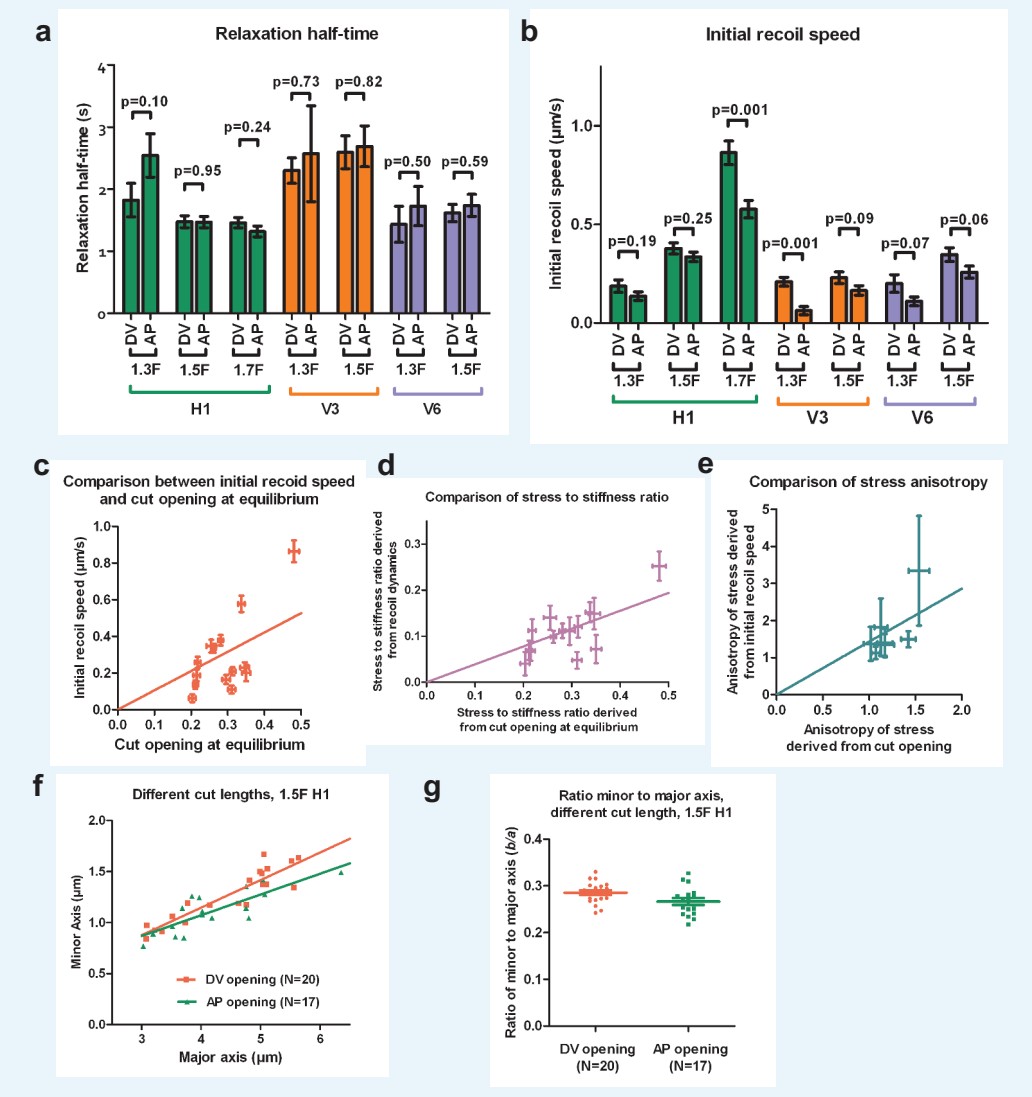

**Appendix 3—figure 1.** Comparison of stress and material properties derived from ablation cut recoil dynamics and opening shape. (**a**) Relaxation half-time and (**b**) initial recoil speed derived from fitting the cut border relaxation using an initial width of cut opening of 0.6 $\mu m$, in H1, V3 and V6 from the 1.3F to the 1.7F stages. Z-test, ns, $p>0.05$; **$0.001<p<0.01$; ***$p<0.001$. DV and AP indicate the directions of opening. (**c**) Comparison between the ratio of stress to stiffness derived using the cut recoil dynamics (after normalization to the cut length) and that derived from the cut opening at equilibrium. (**d**) Comparison of stress to stiffness ratio measured from the recoil dynamics versus that derived using cut opening at equilibrium. (**e**) Comparison of the anisotropy of stress (defined by DV/AP stress) derived from the recoil dynamics and that from the cut opening at equilibrium. (**f**) The minor and major axes of cut opening at equilibrium show a linear relationship when the cut length varied from 3 $\mu m$ to 6 $\mu m$. The ablations were performed in H1 at the 1.5F stage. Solid lines show a linear fit. $R^2 > 0.65$. (**g**) The ratio of the minor to major axis of the cut opening at equilibrium calculated from data shown in (**f**).

## Appendix 4

### The ratio of the minor to major axis of the cut opening at equilibrium does not depend on the cut length

As predicted by *Theocaris et al. (1986)*, the minor to major axis ratio of the cut opening should be independent of the cut length. Indeed, the equation for the major axis $a$ over the cut length $l$ of the opening at equilibrium is given as:

$$\frac{a}{l} = \frac{E + \sigma_{xx} - \sigma_{yy}}{E} \tag{7}$$

where $\sigma_{xx}$ and $\sigma_{yy}$ are the stresses in the principal loading directions $x$ and $y$, respectively, which are also the laser cut directions. $E$ is the Young modulus of the plane (*Appendix 2—figure 1a,b*). Thus, according to the ratio of minor axis to cut length given in *Figure 2b*, the minor to major axis ratio of the ellipse at equilibrium is:

$$\frac{b}{a} = \frac{2\sigma_{yy}}{E + \sigma_{xx} - \sigma_{yy}} \tag{8}$$

and does not depend on the cut length $l$. Our experimental data fitted well with this prediction as shown in *Appendix 3—figure 1f,g*. Indeed, when we plotted the minor versus major axes of the opening at equilibrium, we observed a linear relationship (*Appendix 3—figure 1f*) when the cut length varied from 3 to 6 $\mu m$ in the seam cell H1, for cuts in both the AP and DV directions. Thus, the minor to major axis ratio is nearly a constant and independent of the cut length (*Appendix 3—figure 1g*), consistent with the theory of *Theocaris et al. (1986)*.

## Appendix 5

## Modeling of the *C. elegans* embryo as a capped thin-wall pressured vessel and calculation of the anisotropy of stress

The rationale for modeling the *C. elegans* embryo as a thin-wall pressured vessel is given in the main text. To do so, first, we calculated the stress anisotropy for an axisymmetric vessel. We derived the stress anisotropy for the head from this calculation because the head was considered axisymmetric. Second, we calculated the stress anisotropy in the embryo body at the position of the seam cell V3. The body was not axisymmetric due to the important folding of the embryo in the eggshell, but the stress anisotropy can be obtained using a similar method.

### Anisotropy of stress for an axisymmetric thin-wall pressured vessel

We consider an axisymmetric thin-wall pressured vessel with two ends capped. To make a parallel with *C. elegans* embryos, we call the axis of the vessel AP and the circumferential axis DV (*Appendix 5—figure 1a*). We calculate the anisotropy of stress $\frac{\sigma_{DV}}{\sigma_{AP}}$, where $\sigma_{AP}$ is the longitudinal (AP) stress and $\sigma_{DV}$ is the circumferential stress (DV) on the wall.

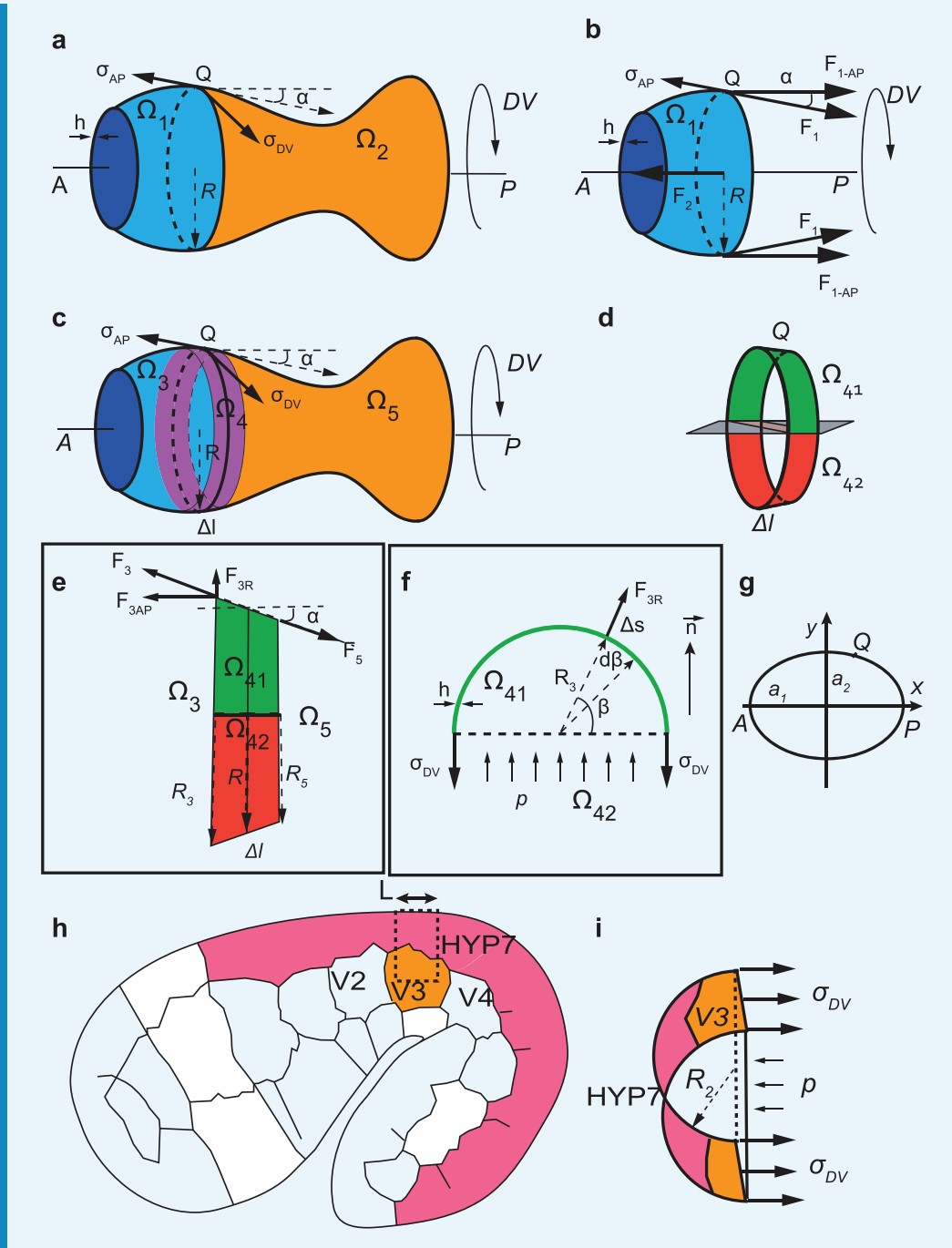

**Appendix 5—figure 1.** Calculation of anisropy of stress for an axisymmetric thin-walled pressured vessel and for *C. elegans* embryos at the seam cell V3 position. (**a-f**) Anisotropy of stress for an axisymmetric thin-wall pressured vessel is derived from the force equilibrium equations of a volume element. (**g**) Geometrical configuration for determining anisotropy of stress for an ellipsoide. (**h-i**) The anisoropy of stress at the level of the seam cell V3 is derived from equilibrium equations of a volume element at V3 postition.

Let us consider a point Q on the wall for which we calculate $\frac{\sigma_{DV}}{\sigma_{AP}}$. The tangent at Q in the plane going through Q and the AP axis makes an angle $\alpha$ with AP.

Imagine that we cut the vessel in two parts by a plane going through Q and perpendicular to the AP axis (**Appendix 5—figure 1a**). The vessel is divided into $\Omega_1$ and $\Omega_2$. The forces applied by $\Omega_2$ on $\Omega_1$ have two components $F_1$ and $F_2$ (**Appendix 5—figure1b**): $F_1$ is the force applied by the wall of $\Omega_2$ on $\Omega_1$, whereas $F_2$ is the force exerted by the hydrostatic pressure from $\Omega_2$. We have

$$F_1 = \sigma_{AP}2\pi Rh \tag{9}$$

where $R$ is the radius of the vessel at Q; $h$ is the thickness of the wall (epidermis), considered as a constant. $F_{1AP}$ is the component in the AP direction of $F_1$ and is written as:

$$F_{1AP} = F_1 cos\alpha = \sigma_{AP}2\pi Rhcos\alpha \tag{10}$$

$F_{1AP}$ is balanced by the hydrostatic force $F_2$ because $\Omega_1$ is at equilibrium

$$F_2 = p\pi R^2 \tag{11}$$

where $p$ is the hydrostatic pressure. By combining the **Equations (10, 11)**, we have:

$$\sigma_{AP}2\pi Rhcos\alpha = p\pi R^2 \tag{12}$$

$$\Rightarrow \sigma_{AP} = \frac{pR}{2hcos\alpha} \tag{13}$$

Now let's consider a volume element $\Omega_4$ of length $\Delta l$ with two limiting sections perpendicular to the AP axis, so that $\Omega_4$ is between $\Omega_3$ and $\Omega_5$ (**Appendix 5—figure 1c**). Imagine that we cut $\Omega_4$ into two halfs $\Omega_{41}$ and $\Omega_{42}$ (**Appendix 5—figure1d,e**).

Let's examine an element $\Delta s$ on the wall of $\Omega_{41}$, at the interface between $\Omega_3$ and $\Omega_{41}$ (**Appendix 5—figure 1f**). The force applied by the wall of $\Omega_3$ to this element is:

$$F_3 = \sigma_{AP}(R_3)h\Delta s = \sigma_{AP}(R_3)hR_3 d\beta \tag{14}$$

where $\sigma_{AP}(R_3)$ means that $\sigma_{AP}$ is a function of $R_3$, $\beta$ is the angle formed by the position of $\Delta s$ with the $\Omega_{41}$-$\Omega_{42}$ interface as shown in **Appendix 5—figure 1f**. The force applied by $\Omega_3$ on $\Omega_{41}$ in the radial direction (**Appendix 5—figure 1e,f**) is:

$$F_{3R} = F_3 sin\alpha(R_3) = \sigma_{AP}(R_3)hR_3 d\beta sin\alpha(R_3) \tag{15}$$

where $\alpha(R_3)$ means that $\alpha$ is a function of $R_3$. The force applied by $\Omega_3$ in the direction $\overrightarrow{n}$ perpendicular to the $\Omega_{41}$–$\Omega_{42}$ interface (**Appendix 5—figure 1f**) is:

$$F_{3n} = \int_0^\pi F_{3R} sin\beta d\beta = \int_0^\pi \sigma_{AP}(R_3)hR_3 sin\alpha(R_3)sin\beta d\beta \tag{16}$$

$$= 2\sigma_{AP}(R_3)hR_3 sin\alpha(R_3) \tag{17}$$

If we replace $\sigma_{AP}$ obtained from the **Equation (13)**, we have:

$$F_{3n} = 2\sigma_{AP}(R_3)hR_3 sin\alpha(R_3) = \frac{pR_3^2 sin\alpha(R_3)}{cos\alpha(R_3)} \tag{18}$$

The force $F_{5n}$ exerted by $\Omega_5$ to $\Omega_3$ can be calculated in the same manner. We obtain:

$$F_{5n} = -\frac{pR_5^2 sin\alpha(R_5)}{cos\alpha(R_5)} \tag{19}$$

The resulting force applied by $\Omega_3$ and $\Omega_5$ to $\Omega_{41}$ in the $\overrightarrow{n}$ direction can be expressed as :

$$F_{35n} = F_{3n} + F_{5n} = \frac{d}{dR}\left(\frac{pR^2 sin\alpha(R)}{cos\alpha(R)}\right)\Delta R = 2pRtan\alpha(R)\Delta R + \frac{pR^2}{cos^2\alpha(R)}\frac{d\alpha}{dR}\Delta R \tag{20}$$

The force applied by $\Omega_{42}$ to $\Omega_{41}$ (**Appendix 5—figure 1e,f**) in the $\overrightarrow{n}$ direction is :

$$F_{12} = p\Delta lcos\alpha(R)2R - \sigma_{DV}2\Delta lh \tag{21}$$

Since $\Omega_4$ is at equilibrium, we have :

$$F_{35n} + F_{12} = 0 \tag{22}$$

Note that

$$\Delta R = \Delta lsin\alpha \tag{23}$$

From the **Equations (20, 21, 22, 23)** we have

$$2pRtan\alpha\Delta lsin\alpha + \frac{pR^2}{cos^2\alpha}\frac{d\alpha}{dR}\Delta lsin\alpha + p\Delta lcos\alpha 2R - \sigma_{DV}2\Delta lh = 0 \tag{24}$$

$$\Rightarrow \sigma_{DV} = \frac{pRsin^2\alpha}{hcos\alpha} + \frac{pRcos\alpha}{h} + \frac{pR^2 sin\alpha}{2hcos^2\alpha}\frac{d\alpha}{dR} = \frac{pR}{hcos\alpha} + \frac{pR^2 sin\alpha}{2hcos^2\alpha}\frac{d\alpha}{dR} \tag{25}$$

Thus we obtain the anisotropy of stress on the wall:

$$AS = \frac{\sigma_{DV}}{\sigma_{AP}} = \frac{\frac{1}{cos\alpha} + \frac{Rsin\alpha}{2cos^2\alpha}\frac{d\alpha}{dR}}{\frac{1}{2cos\alpha}} = 2 + R\frac{d\alpha}{dR}tan\alpha \tag{26}$$

We can now calculate the $AS$ for vessels with a particular shape : a sphere, a cylinder and an ellipsoid.

-—For a sphere of radius $R_0$ :

$$R = R_0 cos\alpha \tag{27}$$

According to the **Equation (13)**, the AP stress is:

$$\sigma_{AP} = \frac{pR_0}{2h} \tag{28}$$

If we take the derivation of the **Equation (27)** with respect to R, we have

$$-R_0 sin\alpha \frac{d\alpha}{dR} = 1 \tag{29}$$

$$\Rightarrow R_0 cos\alpha tan\alpha \frac{d\alpha}{dR} = -1 \tag{30}$$

$$\Rightarrow R tan\alpha \frac{d\alpha}{dR} = -1 \tag{31}$$

Combing the previous equation with the **Equation (26)**, we have:

$$\Rightarrow AS = \frac{\sigma_{DV}}{\sigma_{AP}} = 1 \tag{32}$$

—For a cylinder : $\alpha = const = 0$

$$AS = \frac{\sigma_{DV}}{\sigma_{AP}} = 2 \tag{33}$$

According to the **Equation (13)**, the AP stress is:

$$\sigma_{AP} = \frac{pR}{2h} \tag{34}$$

where $R$ is the radius of the cylinder.

—For an ellipsoid with the major axis $a_1$ and minor axis $a_2$ (**Appendix 5—figure 1g**), we can write the coordinates of point Q in the plane going through Q and AP axis as:

$$\Rightarrow \begin{cases} x = a_1 cost \\ y = a_2 sint \end{cases} \tag{35}$$

thus

$$tan\alpha = \frac{dy}{dx} = \frac{a_2 cost}{a_1(-sint)} = -\frac{a_2}{a_1} cotant \tag{36}$$

$$\frac{d(tan\alpha)}{dt} = (1 + tan^2\alpha)\frac{d\alpha}{dt} = \frac{a_2}{a_1}(1 + cotan^2 t) \tag{37}$$

Thus

$$\frac{d\alpha}{dt} = \frac{\frac{a_2}{a_1}(1 + cotan^2 t)}{1 + \frac{a_2^2}{a_1^2} cotan^2 t} = \frac{a_1 a_2}{a_1^2 sin^2 t + a_2^2 cos^2 t} \tag{38}$$

Due to the symmetry of the system, we examine only $t \in [0, \pi]$. We have:

$$R = y = a_2 sint \tag{39}$$

$$\frac{d\alpha}{dR} = \frac{d\alpha}{dt}\frac{dt}{dR} = \frac{a_1 a_2}{(a_1^2 sin^2 t + a_2^2 cos^2 t)a_2 cost} = \frac{a_1}{(a_1^2 sin^2 t + a_2^2 cos^2 t)cost} \tag{40}$$

From **Equations (36, 39, 40)** we have

$$R\frac{d\alpha}{dR}tan\alpha = \frac{-a_2^2}{a_1^2 sin^2 t + a_2^2 cos^2 t} \tag{41}$$

Thus:

$$AS = \frac{\sigma_{DV}}{\sigma_{AP}} = 2 - \frac{a_2^2}{a_1^2 sin^2 t + a_2^2 cos^2 t} \tag{42}$$

For the middle of the ellipsoid, $y = a_2$ and $x = 0$, $t = \frac{\pi}{2}$, thus

$$AS = \frac{\sigma_{DV}}{\sigma_{AP}} = 2 - \left(\frac{a_2}{a_1}\right)^2 \tag{43}$$

According to the **Equation (13)**, the AP stress at the middle of the ellipsoid is:

$$\sigma_{AP} = \frac{pa_2}{2h} \tag{44}$$

Note that the radial stress on the wall is $-p$. As the wall is thin, i.e $h<<R$, we expect that the radial stress is much smaller than the AP and DV stress on the wall for a sphere, an ellipsoid or a cylinder.

## Anisotropy of stress for the body seam cell V3 of *C. elegans* embryo

For the seam cell $V_3$ at the 1.3F and 1.5F stages, there is an important curvature of the embryo in the ventral part (**Appendix 5—figure 1h**). We cut a part of the embryo going through V3 and the dorsal part of V3 (dash line, **Appendix 5—figure 1h**) and approximate that the resulting half-section as half a cylinder (**Appendix 5—figure 1i**).

The force equilibrium for this part of the embryo in the circumferential direction (**Appendix 5—figure 1i**) is written as:

$$\sigma_{DV}^{V3}2hL = p2R_2L \tag{45}$$

(force generated by circumferential stress on the wall = force due to the hydrostatic pressure), where $\sigma_{DV}^{V3}$ is the DV stress at V3, $h$ is the thickness of the epidermis, $R_2$ is the radius at $V3$, $L$ is the length of the region considered. Thus:

$$\sigma_{DV}^{V3} = \frac{pR_2}{h} \tag{46}$$

For H1 in the head, if we considered the head as a sphere, the DV and the AP stresses are the same and are given as (**Equations 28, 32**):

$$\sigma_{DV}^{H1} = \sigma_{AP}^{H1} = \frac{pR_1}{2h} \tag{47}$$

where $R_1$ is the head radius. If the AP stress is the same for H1 and V3, then the anisotropy of stress at V3 is

$$AS = \frac{\sigma_{DV}^{V3}}{\sigma_{AP}^{H1}} = \frac{\frac{pR_2}{h}}{\frac{pR_1}{2h}} = 2\frac{R_2}{R_1} \tag{48}$$

## Appendix 6

# The Hooke's law written for seam and DV epidermal cells

In this section, we further detail how we used the Hooke's law to describe the deformation of the seam and DV epidermal cells.

## Hooke's law written for seam cells

Hooke's law for the deformation of seam cells is given by:

$$\epsilon_{AP}^s = \frac{\Delta L_{AP}^s}{L_{0AP}^s} = \frac{\sigma_{AP}^s}{E} - \nu \frac{\sigma_{DV}^s}{E} = -\frac{\sigma_{AP}^s}{E}(\nu AS - 1) \tag{49}$$

$$\epsilon_{DV}^s = \frac{\Delta L_{DV}^s}{L_{0DV}^s} = \frac{\sigma_{DV}^s}{E} - \nu \frac{\sigma_{AP}^s}{E} = -\frac{\sigma_{AP}^s}{E}(\nu - AS) \tag{50}$$

Here we supposed that the embryonic cortex material property is isotropic with a Young modulus $E$. $\epsilon_{AP}^s$ and $\epsilon_{DV}^s$ are the strain (which is equal to the relative length change) along the AP and DV directions, respectively; $\Delta L_{AP}^s$ and $\Delta L_{DV}^s$ are the length change, $L_{0AP}^s$ and $L_{0DV}^s$ are the initial length along the AP and DV directions, respectively. Positive values of $\epsilon_{AP}^s$ or $\epsilon_{DV}^s$ correspond to an increase in length (or extension), whereas negative values correspond to a decrease in length (shrinking). $\sigma_{AP}^s$ and $\sigma_{DV}^s$ are the stress along the AP and DV directions, respectively. Positive values of $\sigma_{AP}^s$ or $\sigma_{DV}^s$ correspond to tensile stress, whereas negative values correspond to contractile stress. $\nu$ is the Poisson's ratio describing the shrinking in the AP direction when tensile stress is applied in the DV direction, and vice versa. Here we omit the stress along the radial direction because it is much smaller than the AP and DV stress for a thin-wall vessel (Appendix 5). $AS$ is the stress anisotropy, which equals the DV to AP stress ratio.

$$AS = \frac{\sigma_{DV}^s}{\sigma_{AP}^s} \tag{51}$$

If we have an isotropic spherical embryo covered with contractile seam cells ($AS = 1$), the embryo would not deform due to its incompressibility and symmetry: $\epsilon_{AP}^s = 0$ and $\epsilon_{DV}^s = 0$. From this, we derived that $\nu = 1$. We can thus rewrite *Equations (49 and 50)* as:

$$\epsilon_{AP}^s = -\frac{\sigma_{AP}^s}{E}(AS - 1) \tag{52}$$

$$\epsilon_{DV}^s = -\frac{\sigma_{AP}^s}{E}(1 - AS) \tag{53}$$

## Hooke's law written for DV epidermal cells

The DV epidermal cells have different stiffnesses along the AP and DV directions, so the stress-strain relationship along the AP and DV axes can be written:

$$\epsilon_{AP} = \frac{\Delta L_{AP}}{L_{0AP}} = \frac{\sigma_{AP}}{E_{AP}} - \nu_1 \frac{\sigma_{DV}}{E_{DV}} \tag{54}$$

$$\epsilon_{DV} = \frac{\Delta L_{DV}}{L_{0DV}} = \frac{\sigma_{DV}}{E_{DV}} - \nu_2 \frac{\sigma_{AP}}{E_{AP}} \tag{55}$$

where $\epsilon_{AP}$ and $\epsilon_{DV}$ are the strain (which is equal to the relative length change) along the AP and DV axes, respectively; $\Delta L_{AP}$ and $\Delta L_{DV}$ are the length change, $L_{0AP}$ and $L_{0DV}$ are the initial length of the cell in the the AP and DV directions, respectively. Positive values of $\epsilon_{AP}$ or $\epsilon_{DV}$ correspond to an increase in length (or extension), whereas negative values correspond to a decrease in length (shrinking) of the cells. $\sigma_{AP}$ and $\sigma_{DV}$ are the stress along the AP and DV directions, respectively. Positive values of $\sigma_{AP}$ or $\sigma_{DV}$ correspond to tensile stress, whereas negative values correspond to contractile stress. $\nu_1$ and $\nu_2$ are Poisson's ratios.

Since the head is considered axisymmetric, we have:

$$AS = \frac{\sigma_{DV}^s}{\sigma_{AP}^s} = \frac{\sigma_{DV}}{\sigma_{AP}} \tag{56}$$

and replace

$$\omega = \frac{E_{DV}}{E_{AP}} \tag{57}$$

we have

$$\epsilon_{AP} = \frac{\sigma_{AP}}{E_{AP}} \left( 1 - \nu_1 \frac{AS}{\omega} \right) \tag{58}$$

$$\epsilon_{DV} = \frac{\sigma_{DV}}{E_{DV}} \left( 1 - \nu_2 \frac{\omega}{AS} \right) \tag{59}$$

While the *CB* model seems to be appropriate to describe fiber-reinforced material in extension, its application to describe the shrinking in the fiber direction is questionable. Many fiber-reinforced composites exhibit different stiffnesses in response to extension and compression (**Bert, 1977**; **Jones, 1977**). Furthermore, biological polymers, such as actin filaments, can have different mechanical properties under tensile or compressive stress: actin filaments buckle under compression (**Murrell and Gardel, 2012**). The use of Poisson's ratios for fiber-reinforced material in compression is arguable because the symmetry of the stiffness (or compliance) matrix may not be satisfied (**Bert, 1977**). Given that the DV cells decrease their length in the fiber (DV) direction (**Figure 3e**), the fibers should be under compression. For this reason, we suggest that the DV epidermal cells must have an active mechanism to adjust the actin bundle length to the cell shrinkage along the DV direction, in order to maintain the reinforcement properties.

## Appendix 7

# Cracks opening in orthotropic and fiber-reinforced planes

## Introduction

This appendix gives the proofs of the various relations used in the main paper to extract the residual stresses in the epithelial cells. Exhibiting oriented actin cables, the epithelium can be considered as a thin anisotropic soft layer. The correct description of the epithelium behaviour under strong deformations is achieved via hyper-elasticity. However, the determination of a crack-shape in nonlinear elastostatic remains challenging and has not been achieved to the best of our knowledge. This is why we first assume a linear orthotropic planar material and solve the geometry of the crack by potential functions (**Landau and Lifshitz, 1970**; **Freund and Mechanics, 1990**), following the pioneering contributions of **Muskkhelishvili (1975)**, **Suo (1990)**, **Theocaris et al. (1986)** and **Yoffe (1951)**. In the following, the shape of the crack is given, under simultaneous tension imposed far away along $Ox$ and $Oy$, including also shear stresses (combined Mode $I$ and $II$ of fracture). Then, we present the model for fiber soft material, which is a better representation for living matter. Aiming to estimate residual stresses from the shape aperture, we identify the correspondence between the linear elastic coefficients of anisotropic elasticity and the parameters of a fiber model at low strains.

## Fracture in orthotropic linear elasticity

In material sciences, a common choice of elastic coefficients for orthotropic samples consists of the definition of Young moduli affected to each axis and Poisson ratios defined for each pair of orientation, in addition to shear moduli (equivalent to the second Lamé coefficients [**Landau and Lifshitz, 1970**] $\mu$) . In the case of plane stress elasticity, the equivalent Hooke's law is reduced to six independent coefficients:

$$\begin{cases} u_{xx} &= \frac{1}{E_x}(\sigma_{xx} - \nu_{xy}\sigma_{yy}); \quad u_{yy} = -\frac{\nu_{xy}}{E_x}\sigma_{xx} + \frac{\sigma_{yy}}{E_y} \\ u_{zz} &= -\frac{\nu_{xz}}{E_x}\sigma_{xx} + \frac{\nu_{yz}}{E_y}\sigma_{yy}; \quad u_{xy} = \frac{1}{2\mu_{xy}}\sigma_{xy} \end{cases} \tag{60}$$

The incompressibility condition, $\Sigma_i u_{ii} = 0$, involves the third components of the deformation $u_{zz}$, leading to: $\nu_{xz} = \nu_{yz} = 1$, so finally four elastic independent coefficients are required. When the loads are applied at the border of the sample, the two in-plane components of the equilibrium condition Div $\sigma = 0$:

$$\frac{\partial \sigma_{xx}}{\partial x} + \frac{\partial \sigma_{xy}}{\partial y} = 0 \quad \text{and} \quad \frac{\partial \sigma_{xy}}{\partial x} + \frac{\partial \sigma_{yy}}{\partial y} = 0, \tag{61}$$

are automatically satisfied by the definition of the Airy potential (**Landau and Lifshitz, 1970**): $U(x, y)$:

$$\sigma_{xx} = \frac{\partial^2 U}{\partial y^2}; \quad \sigma_{xy} = -\frac{\partial^2 U}{\partial x \partial y}; \quad \sigma_{yy} = \frac{\partial^2 U}{\partial x^2} \tag{62}$$

Taking into account Hooke's law, **Equation (60)**, one recovers the usual fourth-order partial differential equation for $U$:

$$\frac{\partial^4 U}{\partial x^4} + 2\rho\Lambda^{1/2}\frac{\partial^4 U}{\partial x^2 \partial y^2} + \Lambda\frac{\partial^4 U}{\partial y^4} = 0 \tag{63}$$

where $\Lambda = E_y/E_x$ and $\rho = \frac{1}{2}\sqrt{\Lambda}(E_x/\mu_{xy} - 2\nu_{xy})$. For isotropic materials, $\Lambda = \rho = 1$, because the second Lamé coefficient (related to the Young modulus $E$ and to the Poisson ratio $\nu$) reads: $\mu = E/(2(1+\nu))$. In the isotropic case and in the context of fracture, *Equation (63)* has been solved (*Muskkhelishvili, 1975*) with the help of holomorphic functions and complex analysis, in the case of plane-stress or plain strain elasticity and in Mode *I* (uniaxial loading, perpendicular to the crack direction). The method has been extended to bi-axial loading and arbitrary crack orientation by *Theocaris (1986)* using the same strategy. Coming back to the anisotropic case and as pointed out by *Suo (1990)*, the theoretical analysis differs according to the $\rho$ value. However, to the best of our knowledge, the shape of a crack of finite length has not been determined before, for an orthotropic material. We are concerned with $\rho < 1$, so proofs will be given for $-1 < \rho < 1$ and the results for the shape crack will be simply mentioned without demonstration for arbitrary $\rho$.

## Definition of two complex functions

*Equation (63)* is an even quartic partial differential equation which can be solved by two holomorphic functions $\tilde{F}(z_1)$ and $\tilde{G}(z_2)$ where:

$$z_1 = x + \Lambda^{-1/4}(m + In)y; \quad z_2 = \Lambda^{-1/4}(-m + In)y; \quad n = \sqrt{(1+\rho)/2} \ \text{ and } \ m = \sqrt{(1-\rho)/2}. \quad (64)$$

This treatment is inspired by the work of *Yoffe (1951)* for elasto-dynamic cracks in mode *I* and differs slightly from the work of *Muskkhelishvili (1975)* and *Lekhnitskii (1963)*, which is more fancy but much less intuitive. In the following, the formulation via complex potentials can be checked at each step by elementary calculations. Each stress component $\sigma_{ij}$ also verifies *Equation (63)* and can be written as:

$$\sigma_{yy} = 2Re[F'(z_1) + G'(z_2)]; \quad \sigma_{xy} = -2Re[\mu_1 F'(z_1) + \mu_2 G'(z_2)]; \quad \sigma_{xx} = 2Re[\mu_1^2 F'(z_2) + \mu_2^2 G'(z_2)] \quad (65)$$

where $F(z) = d\tilde{F}(z)/dz$. The reader can check easily that the two components of the equilibrium *Equation (61)* are verified. A standard choice for $F'(z_1)$ and $G'(z_2)$ for a crack lying on the x-axis between $-a < x < a$ is:

$$F'(z_1) = (A_1 + iA_2)\frac{z_1}{\sqrt{z_1^2 - (l/2)^2}} + B_1 + iB_2 \ \text{ and } \ G'(z_2) = (C_1 + iC_2)\frac{z_2}{\sqrt{z_1^2 - (l/2)^2}} + D_1 + iD_2 \quad (66)$$

where the eight constants are real and will be determined by the boundary conditions on the crack lips and the loads far from the crack. On the lips, we have the cancellation of $\sigma_{xy}$ and $\sigma_{yy}$ and both square-roots in *Equation (66)* are imaginary, so it reads:

$$\begin{cases} \sigma_{yy} = 0 \Longrightarrow B_1 = -D_1 \quad \text{and} \quad C_2 = -A_2, \\ \sigma_{xy} = 0 \Longrightarrow C_1 = (C_2 - A_2)m/n - A_1 \quad \text{and} \quad D_2 = (B_1 - D_1)m/n - B_2 \end{cases} \quad (67)$$

Eliminating the solid rotation at infinity leads to $B_1 = -n(mA_2 + nA_1)$, thus giving a fifth relationship. It remains to evaluate the three stress components at infinity $\sigma_{yy}^\infty, \sigma_{xy}^\infty, \sigma_{xx}^\infty$ and we obtain:

$$A_1 = \frac{1}{4}\left(\sigma_{yy}^\infty - \Lambda^{1/4}\sigma_{xy}^\infty/m\right), \quad A_2 = -\frac{n}{4m}\sigma_{yy}^\infty, \quad B_2 = \frac{1}{8mn}\left(-\sqrt{\Lambda}\sigma_{xx}^\infty + \sigma_{yy}^\infty + 2mn^2\Lambda^{1/4}\sigma_{xy}^\infty\right) \quad (68)$$

Finally, taking into account the first relationship of the Hooke's law, *Equation (60)*, knowing that $u_{xx} = \partial u/\partial x$, one can find the horizontal displacement on the lips by quadrature:

$$u_\pm = \frac{2}{\sqrt{E_x E_y}}\{\frac{1}{2}(\sigma_{xx}^\infty \Lambda^{1/2} - \sigma_{yy}^\infty)x \pm n\Lambda^{1/4}\sigma_{xy}^\infty\sqrt{(l/2)^2 - x^2}\} \qquad (69)$$

The determination of the vertical displacement is more subtle because $v$ comes from the shear relation: $\partial v/\partial x = -\partial u/\partial y + \sigma_{xy}/\mu_{xy}$, and we obtain after elimination of solid rotation,

$$v_\pm = \frac{2n}{E_x^{1/4}E_y^{3/4}}\{n\Lambda^{1/4}\sigma_{xy}^\infty x \pm \sigma_{yy}^\infty\sqrt{(l/2)^2 - x^2}\} \qquad (70)$$

In pure tensional loading, $\sigma_{xy}^\infty$, calling $\beta$ the ratio between the imposed vertical tension and the horizontal one: $\beta = \sigma_{xx}/\sigma_{yy}$, we obtain:

$$u_\pm = \Lambda^{1/2}\frac{\sigma_{yy}^\infty}{E_2}(\beta\Lambda^{1/2} - 1)x \quad \text{and} \quad v_\pm = \pm 2n\Lambda^{1/4}\frac{\sigma_{yy}^\infty}{E_y}\sqrt{(l/2)^2 - x^2} \qquad (71)$$

Even if the demonstration given here is for $\rho<1$, a slightly different treatment can be achieved for $\rho = 1$ and $\rho > 1$, but **Equations (69) and (70)** and **Equation (71)** remain valid with the same definition of $n = \sqrt{(1+\rho)/2}$ and $\Lambda$.

Linear elasticity allows us to solve problems of interest exactly in two dimensions, but it is not fully adapted to living matter, which responds differently to low and high stresses. Indeed, in linear elasticity, the material answers linearly to the forcing while for living tissues, we know that large strains resist to the forcing, mostly because of the fibers present in the tissue. Finite elasticity for soft materials is a fast-developing domain but it is technically more difficult. In particular, no exact crack solutions exist. Nevertheless, it presents a better description of the elastic energy: it is the reason why we present hereafter the fiber model in finite elasticity, which is probably more adapted to the epithelium we are considering.

## Fiber model in finite elasticity

For a fibrous material, the elastic energy density $W$ is chosen as the superposition of the energy of a gelatinous matrix $W_{nH}$ (most often, the neo-Hookean or the Monney-Rivlin model [**Ogden et al., 1984**]) and a fiber contribution (**Wu and Ben Amar, 2014**; **Ben Amar et al., 2015**; **Gasser et al., 2006**). Different models exist, which are more or less complicated, based on the experimental responses of fibrous samples to stresses: muscles, arteries, and on required mathematical properties. We select a model which has the property to behave alike in tension or compression, that is the $CB$ model (**Wu and Ben Amar, 2014**; **Ben Amar et al., 2015**), whereas other models, such the $G-O-H$ model (**Gasser et al., 2006**), give a non-symmetric answer in tension and compression. As shown by **Destrade (2015)**, the $CB$ model eliminates unexpected singularities such as that obtained at low strains in dispersion relations. Choosing a neo-Hookean matrix,

$$W_{nH} = \frac{\mu_0}{2}\left(\lambda_1^2 + \lambda_2^2 + \lambda_3^2 - 3\right) \qquad (72)$$

where $W_{nH}$ is function only of the first invariant $I_1 = \text{tr}[\mathbf{F}\mathbf{F}^T]$, ($\mathbf{F}$ being the strain tensor, defined by $F_{ij} = \partial u_i/\partial u_j$), contrary to the Mooney-Rivlin model which also incorporates the second invariant $I_2$. We choose the fiber contribution (**Wu and Ben Amar, 2014**; **Ben Amar et al., 2015**) as:

$$W_{CB} = \frac{\mu_0}{2}K\{2\kappa(\lambda_1^2 + \lambda_2^2 + \lambda_3^2 - 3) + (1 - 3\kappa)(\lambda_2^2 + \frac{1}{\lambda_2^2}) - 2\} \qquad (73)$$

where $\kappa$ is a dispersion coefficient when the fibers are disordered (**Ben Amar et al., 2015**) and $K$ is directly connected to the elasticity of the fibers compared to the elasticity of the matrix. For simplicity, we use $\kappa = 0$ hereafter. As we cannot solve the crack problem for a nonlinear sample, our plan is to consider the low strain limit and to relate the coefficients $\Lambda$ and $\rho$, which are responsible for the crack shape, to the coefficients $\mu_0$ and $K$ of nonlinear elasticity. Considering plane-stress elasticity, minimization of the elastic energy concerns the free energy under the constrain of incompressibility and the condition that the Cauchy stress components $\sigma_{iz}$ cancel.

$$G = \int dX_1 dX_2 dX_3 \{W(\lambda_1, \lambda_2, \lambda_3) - 3) - P\lambda_3 J_{2D}\}, \tag{74}$$

where $J_{2D}$ represents the Jacobian in $2D$ and $P$ is a Lagrange multiplier. Minimization with respect to the strain in the third direction gives:

$$P = \frac{1}{J_{2D}} \frac{\partial W}{\partial \lambda_3} = \lambda_3 \frac{\partial W}{\partial \lambda_3} \tag{75}$$

## Fiber versus orthotropic elasticity in extension

Correspondence between the anisotropic coefficients of linear elasticity and the finite fiber elasticity is possible at small values of strain, such that $\epsilon_i = |\lambda_i - 1| \ll 1$. For plane-stress elasticity, the Cauchy stress leads to:

$$\sigma_i = \lambda_i \frac{\partial W}{\partial \lambda_i} - P = \lambda_i \frac{\partial W}{\partial \lambda_i} - \lambda_3 \frac{\partial W}{\partial \lambda_3} \tag{76}$$

Expanding all $\lambda_i$ for weak deformations, we derive without difficulty, for the C-B model:

$$\epsilon_1 = \frac{1}{\mu_0} \frac{1+K}{3+4K} \left( \sigma_1 - \frac{1}{2(1+K)} \sigma_2 \right) \quad \text{and} \quad \epsilon_2 = \frac{1}{\mu_0} \frac{1}{3+4K} \left( \sigma_2 - \frac{1}{2} \sigma_1 \right) \tag{77}$$

Comparison with the orthotropic Hookean law, **Equation (60)**, in-plane stress elasticity gives:

$$E_x = \mu_0 \left( \frac{3+4K}{1+K} \right); \quad E_y = \mu_0(3+4K); \quad \Lambda = 1+K; \quad \text{and} \quad \nu_{xy} = \frac{1}{2(1+K)} \tag{78}$$

If we call the stiffness of the matrix (without fiber) $E_0$, we have

$$E_0 = 3\mu_0 \tag{79}$$

and

$$E_x = \frac{E_0}{3} \left( \frac{3+4K}{1+K} \right); \quad E_y = \frac{E_0}{3}(3+4K) \tag{80}$$

## Fiber versus orthotropic elasticity in the case of shear deformation

To complete the set of coefficients, we need the shear coefficient $\mu_{xy}$, so we treat a pure shear deformation. Then, the new coordinates in the current deformation are

$$x = X + \Gamma Y; \quad y = Y \quad z = Z \tag{81}$$

giving the deformation tensor $\mathbf{F}$, the left Cauchy-Green tensor (**Ogden et al., 1984**) $\mathbf{F}\mathbf{F}^{\mathbf{T}}$ and the Cauchy stress tensor (**Destrade, 2015**) $\sigma$

$$\mathbf{F} = \begin{bmatrix} 1 & \Gamma & 0 \\ 0 & 1 & 0 \\ 0 & 0 & 1 \end{bmatrix}; \mathbf{F}\mathbf{F}^{\mathbf{T}} = \begin{bmatrix} 1+\Gamma^2 & \Gamma & 0 \\ \Gamma & 1 & 0 \\ 0 & 0 & 1 \end{bmatrix}; \sigma = \begin{bmatrix} 1 & \Gamma\mu_0(1+2K) & 0 \\ \Gamma\mu_0(1+2K) & 0 & 0 \\ 0 & 0 & 0 \end{bmatrix} \tag{82}$$

where $\sigma$ is evaluated according to the following expression (**Destrade, 2015**)

$$\sigma = \mu_0(\mathbf{F}\mathbf{F}^{\mathbf{T}} - \mathbf{I}) + \mu_0 K\{(\mathbf{F} \cdot \mathbf{M}) \otimes (\mathbf{F} \cdot \mathbf{M}) - (\mathbf{F}^{-\mathbf{T}} \cdot \mathbf{M}) \otimes (\mathbf{F}^{-\mathbf{T}} \cdot \mathbf{M})\} \tag{83}$$

restricted to the linear approximation for weak value of the shear strain $\Gamma$. So the shear modulus $\mu_{xy}$ of the orthotropic material is then:

$$\mu_{xy} = \mu_0(1+2K) = E_x\frac{1+K}{3+4K}(1+2K) \tag{84}$$

which allows the calculation of the coefficient $\rho$ introduced in **Equation (63)**:

$$\rho = \sqrt{\Lambda}(\frac{E_1}{2\mu_{xy}} - \nu_{xy}) = \frac{\sqrt{\Lambda}}{2(1+K)}(\frac{3+4K}{1+2K} - 1) = \frac{\sqrt{1+K}}{1+2K} \tag{85}$$

## Evaluation of the residual stress from crack opening

When we cut a fibrous sample perpendicularly to the direction of the fibers, we obviously change the structure locally and also the elastic properties along the crack. We cannot be sure that the *CB* model describes the correct elasticity as the new aperture is free from fibers. Perhaps a better approximation for the shape aperture is an isotropic elasticity. However, far from the crack of length $l$, on a distance larger than $l$, the stresses reach the value at infinity, so $\sigma_{yy}$ and the sample is fibrous. The *CB* model is an approximation as is the isotropic model, and the truth is perhaps somewhere between these extremes. In addition, we consider an infinite sample in all directions $Ox, Oy$, which implicitly assumes that the crack length is small compared to the epithelium size. The the opening ellipse, in the isotropic approximation is then:

$$b_y = 4\frac{\sigma_{yy}}{3\mu_0}\sqrt{(l/2)^2 - x^2} \tag{86}$$

with a crack on the $x$ axis having a length $l$. The minor axis of the opening is then:

$$b_y = 2\frac{\sigma_{yy}}{3\mu_0}l = 2\frac{\sigma_{yy}}{E_0}l \tag{87}$$

Considering now that we cut the sample along the $Oy$ direction, then the cut opening ellipse is

$$b_x = 4n\Lambda^{-1/4}\frac{\sigma_{xx}^\infty(1+K)}{\mu_0(3+4K)}\sqrt{(l/2)^2 - y^2} \tag{88}$$

where $n$ is not modified and varies between $1$ and $1/2$ for increasing stiffness. The minor axis of the opening is:

$$b_x = 2n\Lambda^{-1/4}\frac{\sigma_{xx}^{\infty}(1+K)}{\mu_0(3+4K)}l = 2n\Lambda^{-1/4}\frac{\sigma_{xx}^{\infty}}{E_x}l \tag{89}$$

**Appendix 8**

## Modeling the DV epidermis as an orthotropic material

As outlined in the main text, we could also have modeled the DV epidermis as an orthotropic material (with different stiffnesses in orthogonal directions). In this appendix we argue that doing so is not compatible with the elongation of the embryo.

We modeled the DV epidermis as an orthotropic material with two principal Young moduli along the DV and AP directions, $E_{DV}$ and $E_{AP}$, respectively. We suppose that the stresses $\sigma_{DV}$ and $\sigma_{AP}$ are applied along the DV and AP directions, respectively, and there is no shear stress. The opening in the DV and AP directions is given by **Equation (70)**:

$$\frac{b_{DV}}{l} = 2n \left(\frac{E_{DV}}{E_{AP}}\right)^{\frac{1}{4}} \left(\frac{\sigma_{DV}}{E_{DV}}\right) \tag{90}$$

$$\frac{b_{AP}}{l} = 2n \left(\frac{E_{AP}}{E_{DV}}\right)^{\frac{1}{4}} \left(\frac{\sigma_{AP}}{E_{AP}}\right) \tag{91}$$

where $b_{DV}$ and $b_{AP}$ are the minor axis of the cut opening in the DV and AP directions, respectively; $l$ is the initial cut length. $n$ is the parameter given in the **Equation (64)**.

To obtain the DV/AP Young moduli ratio, if we divide the opening in the DV to the AP direction (same cut length), then we have :

$$\frac{b_{DV}}{b_{AP}} = \left(\frac{E_{AP}}{E_{DV}}\right)^{\frac{1}{2}} \left(\frac{\sigma_{DV}}{\sigma_{AP}}\right) = \left(\frac{E_{AP}}{E_{DV}}\right)^{\frac{1}{2}} AS \tag{92}$$

From the measurement of the AP and DV opening, and given the same anisotropy of stress (AS) in the HYP7 cell as in the H1 cell, we can derive the ratio of DV/AP Young moduli $\frac{E_{DV}}{E_{AP}}$.

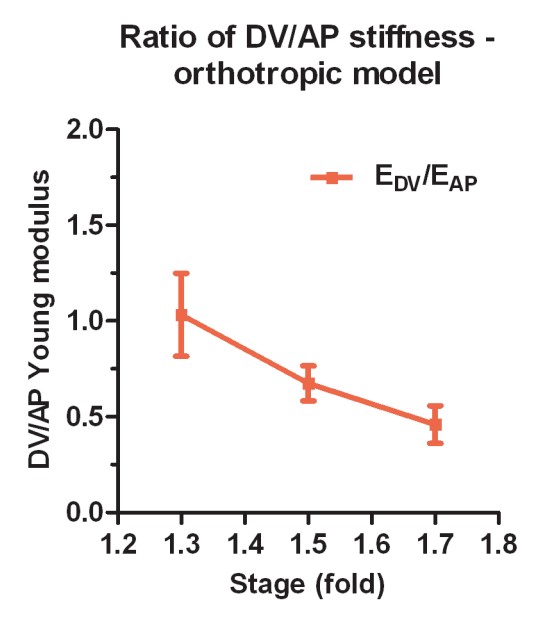

**Appendix 8—figure 1.** Ratio of DV/AP Young moduli calculated from the orthotropic model for

the DV epidermis.

*Appendix 8—figure 1* shows that the ratio of DV/AP Young moduli, calculated from the orthotropic model, decreases as the embryo elongates. This ratio became less than 1 after the 1.3F stage. As the activity of myosin II in the DV epidermis is low, the DV cells are probably submitted to tensile stress from the seam cells. According to Hooke's law written for DV cells (Appendix 6) and given a positive tensile stress on the DV cells, a decrease in the DV/AP Young moduli ratio ($\omega$) should decrease the AP length and increases the DV length. Thus, a decrease in the DV/AP Young moduli ratio as given by the orthotropic model would hinder the elongation of the DV cells in the AP direction and thus the elongation of the embryo as a whole, and would be inconsistent with the contribution of the DV cells during *C. elegans* embryo elongation (*Figure 3e*).

## Appendix 9

### Calculation of the ratio of Young moduli between the seam cell H1 and the head HYP7 cell matrix

To compare the material properties (Young modulus) between the seam cell H1 and the head HYP7 cell matrix (without fibers), we made use of the opening in the DV direction when performing laser cuts in these cells. The DV opening in the seam cell H1 is given as indicated in *Figure 2b*:

$$\frac{b_{DV}^{H1}}{l} = 2\frac{\sigma_{DV}^{H1}}{E} \tag{93}$$

where $b_{DV}^{H1}$ is the minor axis of the cut opening at equilibrium, $l$ is the cut length, $\sigma_{DV}^{H1}$ is the DV stress in H1 and $E$ is the Young modulus of H1. The head HYP7 cell behaved like an isotropic medium with a Young modulus $E_0$ with cuts perpendicular to the actin fibers (DV opening). Thus, the DV opening in the head HYP7 cell is given as:

$$\frac{b_{DV}^{HYP7}}{l} = 2\frac{\sigma_{DV}^{HYP7}}{E_0} \tag{94}$$

where $b_{DV}^{HYP7}$ is the minor axis of the cut opening at equilibrium, $l$ is the cut length, and $\sigma_{DV}^{HYP7}$ is the DV stress in the head HYP7 cell. Given their adjacent position (*Figure 3d*), H1 and head HYP7 should be under the same DV stress:

$$\sigma_{DV}^{H1} = \sigma_{DV}^{HYP7} \tag{95}$$

Thus:

$$\frac{b_{DV}^{HYP7}}{l} = 2\frac{\sigma_{DV}^{HYP7}}{E_0} = 2\frac{\sigma_{DV}^{H1}}{E}\frac{E}{E_0} = \frac{b_{DV}^{H1}}{l}\frac{E}{E_0} \tag{96}$$

When we plotted the DV opening in the head HYP7 cell versus that in H1, the slope gave us the ratio of Young moduli $\frac{E}{E_0}$.

## Appendix 10

### Calculation of K and DV/AP Young moduli ratio for a fiber-reinforced material

For a fiber-reinforced material, from the *Equations (78, 85, 87, 89)*, we have:

$$\frac{b_{DV}}{b_{AP}} = \frac{(3+4K)\sqrt{2}}{3(1+\frac{\sqrt{1+K}}{1+2K})^{\frac{1}{2}}(1+K)^{\frac{3}{4}}} \frac{\sigma_{DV}}{\sigma_{AP}} = \frac{(3+4K)\sqrt{2}}{3(1+\frac{\sqrt{1+K}}{1+2K})^{\frac{1}{2}}(1+K)^{\frac{3}{4}}} AS \tag{97}$$

where $b_{DV}$ and $b_{AP}$ are the minor axis of the DV and AP openings (we used the same cut length $l$), respectively; AS is the anisotropy of stress; and $K$ is the fiber contribution factor. We can measure the openings, so given the anisotropy of stress, we can calculate $K$. We can easily derive the DV/AP Young moduli ratio according to the *Equation (78)*:

$$\omega = \frac{E_{DV}}{E_{AP}} = 1 + K \tag{98}$$

