## [Decision Letter]

[Editors’ note: minor issues and corrections have not been included, so there is not an accompanying Author response.]

Congratulations, we are pleased to inform you that your article, "The interplay of stiffness and force anisotropies drive embryo elongation", has been accepted for publication in eLife. If you have selected our "Publish on Acceptance" option, your PDF will be published within a few days; if you have opted out of the "Publish on Acceptance" option, your work will be published in about four weeks' time. Please take note of the points below and we hope you will continue to support eLife going forwards.

*Reviewer #1:*

This manuscript addresses the mechanobiology of cell and tissue shape changes in the context of *C. elegans* embryonic epidermal elongation. The authors use an impressive combination of laser surgery, live imaging, and mathematical modeling to dissect out the roles of forces and tissue stiffness in elongation. Anisotropies in stiffness in the dorsoventral epidermal cells are found to account for directed elongation in the anterior posterior axis. Without giving a precis of the entire work, this was a very rigorous investigation.

The investigation into stiffness is distinctive and sets this apart from other studies of forces in tissue shape change. From a technical standpoint the biology is complete and very high quality and I have no substantive concerns. The mathematical modeling is beyond my expertise to evaluate. However, I am impressed by the close agreement between the authors' models and their measurements.

*Reviewer #2:*

I read this article with great interest. It is very well written and conducted. It presents an exciting blend of results. To me the most important advance was the demonstration that the mechanical properties of tissue play a significant role in embryonic elongation. The mix of modelling and experimental results brings out most impressive results. The paper should be published as is in my opinion.